# Tropical forcing of increased Southern Ocean climate variability revealed by a 140-year subantarctic temperature reconstruction

Chris S.M. Turney[1,2]*, Christopher J. Fogwill[1,2], Jonathan G. Palmer[1,2], Erik van Sebille[3,4], Zoë Thomas[1,2], Matt McGlone[5], Sarah Richardson[5], Janet M. Wilmshurst[5,6], Pavla Fenwick[7], Violette Zunz[8,9], Hugues Goosse[8], Kerry-Jayne Wilson[10], Lionel Carter[11], Mathew Lipson[1,3], Richard T. Jones[12], Melanie Harsch[13], Graeme Clark[14], Ezequiel Marzinelli[14,15], Tracey Rogers[14], Eleanor Rainsley[16], Laura Ciasto[17], Stephanie Waterman[1,3,18], Elizabeth R. Thomas[19] and Martin Visbeck[20].

[1]Climate Change Research Centre, School of Biological, Earth and Environmental Sciences, University of New South Wales, Australia
[2]Palaeontology, Geobiology and Earth Archives Research Centre, School of Biological, Earth and Environmental Sciences, University of New South Wales, Australia
[3]ARC Centre of Excellence for Climate System Science, University of New South Wales, Australia
[4]Grantham Institute and Department of Physics, Imperial College London, UK
[5]Landcare Research, PO Box 69040, Lincoln 7640, New Zealand
[6]School of Environment, University of Auckland, New Zealand
[7]Gondwana Tree-Ring Laboratory, P.O. Box 14, Little River, Canterbury 7546, New Zealand
[8]Université catholique de Louvain, Earth and Life Institute, Georges Lemaître Centre for Earth and Climate Research, Place Pasteur, 3, 1348 Louvain-la-Neuve, Belgium
[9]Earth System Science and Departement Geografie, Vrije Universiteit Brussels, Belgium
[10]West Coast Penguin Trust, P.O. Box 70, Charleston 7865, West Coast, New Zealand
[11]Antarctic Research Centre, University of Victoria, Wellington, New Zealand
[12]Department of Geography, Exeter University, Devon, EX4 4RJ, UK
[13]Department of Biology, University of Washington, Seattle, Washington, USA
[14]Evolution and Ecology Research Centre, School of Biological, Earth and Environmental Sciences, University of New South Wales, Australia
[15]Sydney Institute of Marine Science, Chowder Bay Rd, Mosman NSW 2088, Australia
[16]Wollongong Isotope Geochronology Laboratory, School of Earth and Environmental Sciences, University of Wollongong, NSW 2522, Australia
[17]Geophysical Institute, University of Bergen, and Bjerknes Centre for Climate Research, Bergen, Norway
[18]Department of Earth, Ocean and Atmospheric Sciences, University of British Columbia, Vancouver, Canada
[19]British Antarctic Survey, Cambridge, UK
[20]GEOMAR Helmholtz Centre for Ocean Research Kiel and Kiel University, Germany

*Correspondence to*: Chris Turney (c.turney@unsw.edu.au)

**Abstract.** Occupying about 14% of the world's surface, the Southern Ocean plays a fundamental role in global climate, ocean circulation, carbon cycling and Antarctic ice-sheet stability. Unfortunately, high interannual variability and a dearth of instrumental observations before the 1950s limits our understanding of how marine-atmosphere-ice domains interact on multi-decadal timescales and the impact of anthropogenic forcing. Here we integrate climate-sensitive tree growth with ocean and atmospheric observations on southwest Pacific subantarctic islands that lie at the boundary of polar and subtropical climates (52-54˚S). Our annually-resolved temperature reconstruction captures regional change since the 1870s and demonstrates a significant increase in variability from the 1940s, a phenomenon predating the observational record. Climate reanalysis and modelling show a parallel change in tropical Pacific sea surface temperatures that generate an atmospheric Rossby wave train which propagates across a large part of the Southern Hemisphere during the austral spring and summer. Our results suggest modern observed high interannual variability was established across the mid-twentieth century, and that the influence of contemporary equatorial Pacific temperatures may now be a permanent feature across the mid to high latitudes.

## 1 Introduction

Observations during the second half of the twentieth century suggest significant but spatially complex variability in atmospheric and ocean temperature and circulation (Figs. 1 and S1), as well as ice-sheet dynamics, across the mid- to high-latitudes of the Southern Hemisphere (Jones et al., 2016). These factors include an intensification of western boundary currents (Wu et al., 2012), a strengthening and poleward shift in the summer westerly winds associated with a positive trend in the Southern Annular Mode (SAM) (Marshall, 2003; Abram et al., 2014; Thompson et al., 2011), winter-spring warming over West Antarctica (Steig et al., 2009), latitudinal shifts in the Subantarctic and Polar fronts associated with the Antarctic Circumpolar Current (ACC) (Langlais et al., 2015), spatial and temporal changes in sea ice extent (Turner et al., 2015; Hobbs et al., 2016), and Antarctic ice-sheet mass loss (Pritchard et al., 2012). Unfortunately, major uncertainties exist regarding their trends and interaction(s) due to high interannual variability (Turner et al., 2016; Fogt et al., 2012) and limited instrumental records prior to the 1950s (Goosse and Zunz, 2014; Jones et al., 2016). As a result, analysis has relied on modelling studies to infer multi-decadal to centennial variability (Freitas et al., 2015; Wang and Dommenget, 2015) and explore regional and global teleconnections (Langlais et al., 2015; Goosse and Zunz, 2014), both of which may have changed with anthropogenic forcing. The above uncertainties are particularly acute in the south Pacific Ocean and adjoining regions because of the expression of central tropical ocean-atmospheric interactions associated with the El Niño-Southern Oscillation (ENSO) (Abram et al., 2014; Schneider et al., 2012; Turney et al., 2016a; Ciasto and Thompson, 2008; Ding et al., 2012).

Late twentieth century climate over the Southern Ocean is characterised by high inter-annual variability (Jones et al., 2016; Turner et al., 2016), driven mainly by changes in the strength and location of mid-latitude westerly airflow (Thompson et al., 2011). SAM and ENSO play a dominant role in this as modes of large scale variability (Fogt et al., 2012; Ciasto and Thompson, 2008). Of particular significance, the positive post-1960s trend in the mid-to-high latitude pressure gradient described by SAM reaches its maximum during the austral summer (Jones et al., 2016), marked by a zonally symmetric poleward displacement of the jet stream and

strengthening of the prevailing surface westerly air flow centred on 50°S (Marshall, 2003; Thompson et al., 2011) (Fig. S2). In contrast, ENSO is associated with spatially different temperature and wind relationships across mid to high latitudes (Ciasto and Thompson, 2008) (Fig. S3), with atmospheric pressure anomalies experiencing their greatest amplitude during austral spring and summer in the south Pacific (Fig. S4). The pattern resembles a zonally asymmetric wave train of atmospheric pressure anomalies extending from New Zealand to the west Antarctic coast, and into the Weddell Sea–South Atlantic (the so-called Pacific-South American or PSA mode) (Mo and Higgins, 1998; Trenberth et al., 2014; Trenberth et al., 1998; Karoly, 1989). The PSA has been shown to introduce zonal asymmetries in the seasonal SAM structure in the south Pacific (Fogt et al., 2012). Overall, the poleward migration of storm tracks reduces air-to-sea heat fluxes through increased cloud cover and evaporative heat loss from the ocean (Thompson et al., 2011; Ciasto and Thompson, 2008), while increasing oceanic Ekman transport of cool surface water (Ciasto and Thompson, 2008) and a poleward eddy heat flux (Sallée et al., 2012). As a result, sector-specific poleward shifts in westerly airflow have led to contrasting late twentieth century ocean-atmospheric trends. How the above modes of variability influenced Southern Ocean climate and ocean dynamics before the period of satellite observations remains highly uncertain (Jones et al., 2016). An improved network of quantified climate-sensitive proxy records across the mid- to high-latitudes is crucial for exploring climate teleconnections through time (Jones et al., 2016; Abram et al., 2014; Turney et al., 2016a; Turney et al., 2015).

The subantarctic islands of the southwest Pacific lie at hemispherically important atmospheric and ocean boundaries, offering considerable potential for understanding long-term climate trends and the potential role of tropical forcing on high latitude change. Campbell (52.54˚S 169.14˚E) and Macquarie (54.50˚S 158.95˚E) islands are located just north of the main front of the ACC and south of the Subtropical Front (also known as the Subtropical Frontal Zone or Convergence) (Fig. S6) (Streten, 1988; Sokolov and Rintoul, 2009) in the core latitude of Southern Hemisphere westerly airflow (Streten, 1988), and are sensitive to Rossby wave propagation from the tropics to the high-latitudes (Adamson et al., 1988; Ding et al., 2012) (Fig. S4). Campbell and Macquarie islands have some of the longest, near-complete, continuous instrumental records in the Southern Ocean (commencing 1941 and 1948 respectively) (Table S1) supplemented by daily atmospheric and sea surface temperature (SST) measurements made at Macquarie Island between Common Era (CE) 1912 and 1915 as part of Sir Douglas Mawson's landmark Australasian Antarctic Expedition (AAE) (Kidson, 1946). Mawson's observations span four years and resolve the seasonal cycle, therefore allowing comparison to the continuous instrumental record from the 1940s to present day (hereafter 'the modern record'). The time series of observed temperatures on the two islands are highly correlated in the modern record (detrended July-June correlation 0.801, $p < 0.0001$) and display the same significant spatial correlation fields to regional and Pacific-wide SSTs (Fig. S5), demonstrating a comparable climate regime. As a result of anomalies in the overlying wind, the surrounding waters are strongly influenced by variations in northward Ekman transport of cold fresh subantarctic surface water and anomalous fluxes of sensible and latent heat at the atmosphere–ocean interface. This has produced a cooling trend since 1979 (Figs. 1 and S1) (Thompson et al., 2011; Ciasto and Thompson, 2008), making the islands ideally placed to detect wind-driven changes in the north-south SST gradient over time.

Here we extend the instrumental record by exploiting the climate sensitivity of the southernmost growing trees
in the subantarctic southwest Pacific to produce the first annually-resolved quantified temperature
reconstruction for the region back to CE 1870. The maritime climates of Campbell and Macquarie islands
provide a reconstruction of air and sea temperatures that demonstrates increasing variance since modern records
commenced in the ~1940s. We investigate this time series with climate reanalysis and a three-dimensional
Earth-system model of intermediate complexity and identify the tropical Pacific sea surface temperatures as the
principal driver of the observed variance, propagated by atmospheric Rossby waves during the austral spring
and summer. Climate-sensitive records across the circum-Pacific demonstrate comparable trends, suggesting
that tropical climate changes have been increasingly projected onto the mid to high latitudes. Subantarctic
islands across the wider Southern Ocean provide crucially situated landmasses from which proxy data can be
generated to test hypotheses about past and future global climate teleconnections.

**2 Methods**
**2.1 Subantarctic island climate datasets (Macquarie and Campbell Islands)**
The Australasian Antarctic Expedition (AAE) 1912-1915 atmospheric observations were taken from the isthmus
at the northern end of Macquarie Island (Newman, 1929), in the same immediate area as the current Australian
Antarctic Division Meteorological Station, established in late 1948 (54.50˚S 158.95˚E). The daily and monthly
meteorological data from Macquarie Island were obtained from the reduced AAE dataset (Newman, 1929) and
since 1948, the Bureau of Meteorology (http://www.bom.gov.au/climate/data-services/). Twice-daily SST
measurements were also taken from Buckles Bay during the AAE (Newman, 1929), with subsequent
observations made again during the 1950s and 1960s (Loewe, 1968, 1957); unfortunately no direct
measurements exist between 1916-1950. The next available continuous SST observations that can be compared
to the 1912-1915 record are remote MODIS satellite measurements providing 4 km-resolved 11 μm daytime
observations since 2001 (data accessed via http://gdata1.sci.gsfc.nasa.gov/); other satellite products do not
resolve a spatial scale that allows a direct comparison to the localised measurements made at Macquarie Island.
Although the satellite data are from a larger area than the AAE observations, the expedition vessel the S.Y.
*Aurora* made SST measurements across Buckles Bay and demonstrated similar absolute values as those
observed inshore, providing confidence that the comparisons are robust (Kidson, 1946). A meteorological
station has operated in Perseverance Harbour, Campbell Island (52.54˚S 169.14˚E), since 1941. The dataset
used here was obtained from the New Zealand National Climate Database (http://cliflo.niwa.co.nz/). Near
complete instrumental records have been maintained on Campbell and Macquarie islands since observations
began with no complete months missing from any of the datasets (Table S1).
**2.2 Meteorological observations**
To extend the satellite record for the southwest Pacific, we focused on the subantarctic Macquarie and Campbell
islands. For comparison to the AAE 1912-1915 record, modern-day Macquarie Island temperature
measurements were compared in four-year bins (Tables S2 and S3). The interannual variability in the most
complete dataset (that from Macquarie Island) is relatively large. Student's t-tests (two-tailed) of the four-year
average monthly data relative to 1912-1915 indicate that the most consistently warmer conditions are during
February-April (Tables S2 and S3). This analysis illustrates a trend towards seasonally-restricted warming only
during the late austral summer and autumn. Intriguingly, no pervasive warming is observed across the austral
spring and most of the summer when ENSO and SAM are today known to play a dominant role on regional
climate variability (Ciasto and Thompson, 2008).
**2.3 Tree-ring reconstruction (dendrochronology)**
To develop an annually-resolved temperature reconstruction for the southwest Pacific that will extend the
modern instrumental record we sampled 30 *Dracophyllum* spp. trees from Campbell Island during 2013 as part
of the Australasian Antarctic Expedition 2013-2014, and during further fieldwork in late 2014 (Fig. 2). Here two
*Dracophyllum* species (*D. longifolium*, *D. scoparium* and hybrids) form the southernmost growing evergreen
shrubs and small trees in the southwest Pacific (with no *Dracophyllum* on Macquarie Island) (Wilmshurst et al.,
2004; Turney et al., 2016b). *Dracophyllum* spp. are known to be responsive to warmer temperatures and capable
of reaching ages of >200 years (Harsch et al., 2014), providing an opportunity to derive a continuous proxy
record of temperature in this key region spanning more than a century. Because of the coherent climate trends
on both islands, the relationship of tree-ring growth to Campbell and Macquarie Islands temperature records was
explored using bootstrapped correlation function analysis in the bootRes R software package (Zang and Biondi,
2012) to identify the monthly temperature responses, followed by a split-period for calibration/verification
analysis to test the regression model robustness using the reduction of error (RE) and the coefficient of
efficiency (CE) (Fig. 2 and Table S4). Based on those results, we selected an austral 'growing season' window
for linear regression modelling to produce spring-summer (October-March) temperature reconstructions for
Campbell and Macquarie islands.

After crossdating and measuring, the 30 tree-series were standardised to remove biological trends using the
RCSigFree program (www.ldeo.columbia.edu/tree-ring-laboratory/resources/software).  Within the program,
various options are available for the conversion of the annual ring-width measurements into indices and we
adopted the use of a more flexible regression model, the Friedman's Super Smoother (Friedman, 1984), to
remove the growth trends.  The ring-width measurements were first power transformed and then subtracted from
the regression model to produce indices and avoid possible outlier bias (Cook and Peters, 1997). Following this,
the signal-free method was applied to minimise trend distortion and end-effect biases in the final chronology
(Fig. 2) (Melvin and Briffa, 2008).  Comparison between the detrended series and average raw measurements
(Fig. 2) demonstrate the standardisation process (or any of the other models) did not make the series
heteroscedastic. The relationship of the tree-ring chronology to the Campbell and Macquarie Islands
temperature records and Southern Annular Mode (SAM) reconstruction (Visbeck, 2009) was explored using
bootstrapped correlation function analysis in the bootRes R software package (Figs. 2 and S17) (Zang and
Biondi, 2012). bootRes uses 1000 bootstrapped samples to compute Pearson's correlation coefficients between
the tree-ring parameter and each of the climatic predictors and then to test their significance at the 0.05 level.
Bootstrap samples are drawn at random with replacement from the selected time interval. Median correlation
coefficients are deemed significant if they exceed, in absolute value, half the difference between the 97.5th
quantile and the 2.5th quantile of the 1000 estimates (Biondi and Waikul, 2003).  In the plots the darker bars
indicate a coefficient significant at $p < 0.05$ and the lines represent the 95%-confidence interval.

Based on these results, the Campbell Island "growing-season" of monthly temperatures from October to March
(six months, spanning from spring to autumn) was selected for reconstruction using the *Dracophyllum*
chronology for the period 1870-2013 (Expressed Population Signal or EPS>0.85). Similarly for Macquarie
Island, the same growing season was selected (October-March; six months).  For SAM, we find the most
significant relationship was for July-October. The program PCReg (www.ldeo.columbia.edu/tree-ring-
laboratory/resources/software) was used to carry out a linear regression model of the tree-ring chronology to the
selected growing-season windows for both Campbell and Macquarie Islands. A split-period for
calibration/verification analysis was used (Cook and Kairiukstis, 1990) to test the regression model robustness.
Our model for Campbell Island passed both the CE and RE tests (i.e. positive) indicating that the model was
skillful in reconstructing observed variations, however the verification results for Macquarie Island were weaker
and just failed for the more rigorous CE test (Table S4). We then used the full period of instrumental data
(1949-2012 for Campbell Island and Macquarie Island) to develop final models and reconstruct "growing-
season" temperatures back to 1870 for both islands (Fig. 3).  The prediction intervals (90% quantile limits)
associated with the reconstructed temperatures were produced using a fixed *t*-statistic for scaling the
uncertainties (Olive, 2007). Importantly, the chronology is derived from a mixture of tree ages (i.e. the oldest
started in CE 1747 and the youngest in 1958) and is not made up of a single cohort of similar aged trees that
have matured across the same period.

**2.4 Spectral Analysis**
To investigate climate periodicities we undertook multi-taper method (MTM) analysis on the *Dracophyllum*
temperature reconstruction (Fig. 4), tree chronology (Fig. S12) and annual southwest Pacific SSTs (Fig. S13)
(the latter derived from Hadley Centre Ice and Sea Surface Temperature; HadISST) (Rayner et al., 2003) using a
narrowband signal and red noise significance (with a resolution of 2 and 3 tapers) (Thomson, 1982) with the
software *kSpectra* version 3.4.3 (3.4.5).

**2.5 Characterising water mass sources and ocean fronts**
In order to characterise the decadally-averaged source(s) of water masses near Macquarie and Campbell islands,
we performed an experiment with virtual particles in an eddy-resolving ocean model (the Japanese Ocean model
For the Earth Simulator or OFES) (Masumoto et al., 2004), which has a 1/10° horizontal resolution and near-
global coverage between 75°S and 75°N, and has a demonstrated ability for modeling changes in the Southern
Ocean between 2000 and 2010 (van Sebille et al., 2012) (Fig. S6). While OFES precludes us from modelling the
warming across the 1970s, it does allow us to hindcast the origin of the waters down to a depth of 400 metres
using Lagrangian analysis in the most-recent decade. Assuming a steady-state ocean circulation, this analysis
allows us to refine our understanding of the sources and by association boundaries of water masses surrounding
Macquarie and Campbell islands. The model was forced using the National Centers for Environmental
Prediction (NCEP) wind and flux fields and output is available as three-day averages (Qin et al., 2014).
Particles were released every three days between 1 January 2005 and 31 December 2010 on a latitude-depth
section at 170°E, every 0.1° in latitude between 60°S and 45°S and every 50 m in depth between 25 m and 300
m, for a total of 318,288 particles. The particles were then advected backwards in time within the three-
dimensional OFES velocity fields using the fourth-order Runge-Kutta method as implemented in the
Connectivity Modeling System (CMS) version 1.1b (Paris et al., 2013). The particles were advected for five
years, or until they reached 30°S or 0°E. Once all the particles were integrated, they were categorised into those
that start in the Agulhas Current (at 30˚S and between 28˚E and 40˚E) and those that start in the East Australian
Current (at 30˚S and between 150˚E and 160˚E).
Using the western Indian Ocean boundary Agulhas Current as a tracer for the Subtropical Front (and the
southern limit of the Subtropical Gyre) (Wang et al., 2014) we identify a pathway of particles flowing from the
Cape of Good Hope to the southwest Pacific subantarctic islands (Fig. S6). The particles that connect to the
region around Macquarie and Campbell islands follow a very narrow and almost linear path southeastward
across Indian Sector of the Southern Ocean. The fastest particles reach Macquarie Island less than two years
after release in the Agulhas Current, with the majority arriving between three and four years after release. In
contrast, little leakage from the East Australian Current is observed, with approximately six times more Agulhas
particles delivered to the southwest Pacific subantarctics than the East Australian Current (EAC) (Wu et al.,
2012) (Fig. S6).

**2.6 Modelling transient change**
General circulation models (GCMs) involved in the Fifth Coupled Model Intercomparison Project (CMIP5)
(Taylor et al., 2011) struggle to simulate the observed internal variability and/or seasonal cycle over the
Southern Ocean (Wang et al., 2015; Zunz et al., 2013), supported by the poor correlations observed between our
reconstructed and CMIP5 October-March temperatures (Table S5). Here we take an alternative approach using
LOVECLIM1.3, a three-dimensional Earth-system model of intermediate complexity (Goosse et al., 2010) that
includes representations of the ocean and sea ice (CLIO3) (Goosse and Fichefet, 1999), atmosphere (ECBilt2)
(Opsteegh et al., 1998) and vegetation (VECODE) (Brovkin et al., 2002). The three-level quasi-geostrophic
atmospheric model has a horizontal resolution approximating 5.6˚ × 5.6˚ (T21) whilst the ocean general
circulation model is coupled to a sea-ice model with 20 unevenly spaced vertical levels and a horizontal
resolution of 3˚×3˚. The vegetation component simulates the evolution of grasses, trees and desert, with the
same horizontal resolution as ECBilt2. The experiments analysed here cover the period CE 1850– 2009, driven
by the same natural (solar and volcanic) and anthropogenic (greenhouse gas, sulfate aerosols, land use) forcings
(Goosse et al., 2006) as the ones adopted in the historical simulations performed in the framework of CMIP5
(Taylor et al., 2011). The initial conditions are derived from a numerical experiment covering the years CE 1–
1850 using the same forcing, in order to take into account the long memory of the Southern Ocean (Goosse and
Renssen, 2005). For the CE 1850–2009 simulations, the model was forced to follow the observations of surface
temperature from the HadCRUT3 dataset (Brohan et al., 2006) using a data assimilation technique based on
particle filtering (Goosse et al., 2006; Dubinkina and Goosse, 2013). A simulation without additional freshwater
flux (no freshwater flux) with data assimilation, from CE 1850 to 2009, was analyzed here (Zunz and Goosse,
2015), allowing direct comparison between climate parameters and SST trends across the Southern Ocean. SSTs
for the Macquarie-Campbell island sector and anomalies in zonal wind stress.
**3 Results and Discussion**
**3.1 Modern climate changes**
Comparing atmospheric temperatures during the 1912-1915 AAE observational period and the modern record
from Macquarie Island demonstrates high interannual variability (Fig. 1B). Whilst the temperature trend across
the period of satellite observations appears to show a cooling trend in the southwest Pacific, significant warming
is observed across the annual and spring-summer months from the 1960s and peaks during the 1980s (Figs. 1
and S7, Table S2). No parallel changes are observed in wind direction (Fig. S8) while the sunshine time series
appears to trend in the opposite direction to that expected (Fig. S9). The number of ocean observations are more
limited, but comparable warming (0.5°C) was observed across the 1950s-1960s with MODIS satellite
measurements (MODerate Imaging Spectroradiometer; 2000-2014) demonstrating slightly cooler waters during
the present day (though still 0.3°C warmer than the AAE period) (Table S3). A similar long-term trend is also
observed with air and sea temperatures at Campbell Island (Morrison et al., 2015). Importantly, because of their
small size and highly maritime climate, atmospheric temperatures on the islands parallel the seasonal SST cycle
(Fig. 1C), indicating a tight thermal coupling between air and sea surface temperatures (Thompson et al., 2011;
Kidson, 1946; McGlone et al., 2010), providing a sensitive terrestrial measure of Southern Ocean conditions.

**3.2 Changing climate variability**
The *Dracophyllum* reconstructions extend the surface air temperature record for the southwest Pacific sector of
the Southern Ocean back to CE 1870 (Fig. 3). We find highly-variable growing season (spring-summer)
temperatures that parallel meteorological observations on the subantarctic islands for the period of overlap
(including the original AAE) (Fig. 3), with a trend towards increasing temperatures from the 1960s that reached
a maximum during the late 1980s (~1°C warmer on Macquarie Island compared to period 1912-1915). Peak
temperatures of the 1980s, however, were not sustained in the southwest Pacific through to present day (Fig. 1).
Instead, a notable feature of our 140-year reconstruction is the long-term change in variability captured by a 30-
year running standard deviation, regardless of the standardisation method used (Figs. 3C and S11). We observe
a sustained increase from the ~1940s compared to intermediate levels of variance during the late nineteenth
century and a minimum during the first half of the twentieth century. The high number of replicated trees across
the reported series means we can discount changing sample depth as the cause of increasing variance.
Removing extreme values centred on 1956, 1979 and 1986 does not substantially change the shift to higher
variance in the second half of the twentieth century (Fig. S10), demonstrating that the long-term trend is robust.
To test for the significance of this change, we compared the variance across the tree-ring record (CE 1870-1941
vs 1941-2012) and found the second half of the twentieth century is significantly larger for all standardization
approaches (Friedman F and Bartlett's K-squared tests *p* = 0.0055; Table S6), suggesting a shift in climate to
one characterised by pervasive high variability.
To further investigate the change in temperatures across the record we undertook multi-taper method (MTM)
spectral analysis on the reconstructed air temperature and associated tree-ring index (Figs. 4 and S12). We find
the strongest periodicities in growing season temperatures over two narrow windows, 3 and 2.4 years (all above
95% confidence), identical to those recognized in regional SSTs extracted from HadISST (Rayner et al., 2003)
(Fig. S13). Hovmöller plots of satellite-observed SSTs between 45˚ and 55˚S confirm a pattern of alternating
warm and cool temperatures in the southwest Pacific subantarctic islands with these periodicities (Fig. S14).
Our new extended temperature series therefore indicates the late nineteenth and early twentieth century climate
was characterised by low inter-annual variability with increasing amplitude in the 3 and 2.4-year bands from the
~1940s and late 1960s respectively (Fig. 4B). Recent work by Chelton and Risien (2016) suggest that there is an
increase in standard deviation in HadISST from 1949. Our tree-ring temperature reconstruction, however, shows
a real variance increase that is independent of this artifact in the observational data. We therefore conclude the
increased amplitude of the 3 and 2.4-year bands is a robust climate feature in the southwest Pacific since the
1940s.
**3.3 Marine population changes**
Recent work has illustrated how multi-stressors (including climate variability) can impact on Southern Ocean
biota (Boyd et al., 2015) and have potentially dramatic biological responses across different trophic levels
(Trathan et al., 2007; Constable et al., 2014), including reduced breeding success (Lea et al., 2006). Intriguingly,
the observed increase in variance reported here appears to coincide with a regional order of magnitude decline
in the populations of many marine species across the southwest Pacific (spanning Macquarie Island to the
Antipodes Islands), including penguins and elephant seals (Weimerskirch et al., 2003; Morrison et al., 2015;
Childerhouse et al., 2015; Moore et al., 2001; Baker et al., 2010). Top marine predators can provide an
integrated view of an ecological system, offering a measure of the impact of climate changes on the availability
of food supplies (abundance and distribution), and on feeding and breeding habitats (Jenouvrier et al., 2003).
Whilst not a focus of the current study, the following provides a brief summary of penguin and elephant seal
population trends as a basis for comparison to the climate and ocean trends and variability reported here.
In the New Zealand subantarctic there have been pronounced declines in the numbers of Eastern Rockhopper
penguins (*Eudyptes filholi*) at Campbell Island, and both Rockhopper and Erect crested penguins (*E. sclateri*) on
the Antipodes Islands (49.68˚S 178.75˚E) (Table S7). On Campbell Island, the 1940s breeding population of
Rockhopper penguins was estimated at 1.6 million birds, declining through the 1950s followed by a brief
resurgence in numbers, before a further decline that began no later than the mid-1970s (Cunningham and Moors,
1994). By 2012, Rockhopper numbers on Campbell Island had suffered a 95.5% decline (of which 94% had
occurred by the mid-1980s) (Morrison et al., 2015). Allowing for a lag of several years for chicks to reach
breeding age, the changes in Rockhopper penguin numbers correlate with changes in sea water temperatures
recorded in Perseverance Harbour which increased to a peak between 1945 and 1950, declined between 1950
and 1965, then increased sharply by 1970 (Morrison et al., 2015; Cunningham and Moors, 1994). For the

Antipodes, data on the decline in both Eastern Rockhopper and Erect crested penguin populations cover a shorter period, but are more robust. Whole island group surveys have been conducted on three occasions and, although there were some differences in counting methodology and time of year in which counts were made, the decline in both species has been substantial; in 2011 there were only about 5% as many Rockhopper penguins and fewer than half as many Erect crested penguins as there were in 1978 (Table S7) (Hiscock and Chilvers, 2014). Whilst no climate data are available from the Antipodes Islands, this subantarctic archipelago falls within the same climate zone as Macquarie and Campbell Islands (Fig. 1) and is therefore assumed to have experienced the same long-term trend in air and sea surface temperatures.

Land-based threats do not account for the declines observed. Nesting habitat availability is unchanged and introduced mammals are not generally considered to pose a threat. On Campbell Island, Norway rats (*Rattus norvegicus*) and feral cats (*Felis catus*) were present until eradicated in 2001. However, rats are thought to only prey on eggs once they are broken through other causes and there was no evidence to suggest that the few cats present preyed on Rockhopper penguins, their eggs or chicks (Cunningham and Moors, 1994). Avian cholera was recorded in Campbell Island Rockhopper penguins in 1885/86 and 1986/87, but the numbers killed do not account for the magnitude of the declines recorded (Cunningham and Moors, 1994). Feral sheep (*Ovis aries*) were present (since eradicated) but penguin numbers declined in both accessible and inaccessible colonies (Cunningham and Moors, 1994). On Antipodes Islands, House mice (*Mus musculus*) are the only introduced mammal and they are too small to pose a threat to penguins.

Similar to penguin populations, the number of elephant seals (*Mirounga leonina*) have also declined on both Campbell and Macquarie Islands since the 1940's with the decrease being most marked on Campbell island which is further from the Polar Front (Antarctic Convergence), considered to be the optimum foraging habitat for the species (Taylor and Taylor, 1989). Pup production on Campbell Island declined from 191 individuals in 1947, 11 in 1984, to just five in 1986 (Taylor and Taylor, 1989). On Macquarie Island, the numbers fell 45-55% between the 1950's and 1985 (Hindell and Burton, 1987). The most likely explanation for those declines are decreases in marine food availability due to changes in the marine environment.

Because of the scarcity of island breeding sites and their limited foraging range while breeding, subantarctic penguins are particularly susceptible to climate change and associated changes in marine parameters. Penguins, elephant seals and other top predators, may respond to changes in food availability when marine parameters change by retracting or expanding their distributions, with changes in population size or breeding phenology (Weimerskirch et al., 2003). Alternatively, climate change can affect populations due to changes in conditions ashore. For example, at Punta Tombo in Argentina since 1960 storms have become more frequent and more intense causing the deaths of Humboldt penguin (*Spheniscus humboldti*) chicks (Boersma and Rebstock, 2014). At Punta Tombo, chick deaths due to storms were additive to deaths due to other factors. It is important to note, however, that there is usually a lag between climate change and any subsequent change in penguin (or other predator) population; the lag time depending on whether climate affects adult or chick survival, recruitment or some other demographic parameter (Weimerskirch et al., 2003). Future work is now needed to investigate this relationship further and identify which changes in marine parameters may be the cause.

**3.4 Investigating ocean-atmosphere teleconnections**

Whilst the southwest Pacific subantarctic islands lie along the northern edge of the ACC and south of the Subtropical Front (Fig. S6), the absence of propagating SST signals across the Southern Ocean suggests that movement of ocean boundaries and/or changing input of marine western boundary currents (Figs. S6 and S14) are not primary drivers of the observed increased variability. An alternative scenario for the increasing amplitude in the 3 and 2.4-year bands is a change in atmospheric circulation. To identify a possible atmospheric mechanism, we compared air temperatures over Macquarie Island with estimates from ERA Interim reanalysis (Dee et al., 2011) and observe a significant positive correlation to spring-summer atmospheric pressure anomalies (deseasonalised and detrended at 850 hPa) since 1979 (Fig. 5A) and inverse relationships with temperature and zonal and meridional wind stress (Figs. 5B and S15). Cooler temperatures over Macquarie Island are therefore associated with a centre of relatively low pressure (at 850 hPa) south of New Zealand and enhanced westerly and southerly airflow across a longitudinal band spanning 120˚ to 150˚E (significance $p_{field}$ < 0.05). A similar positive correlation to spring-summer SSTs is observed with both Macquarie Island (Fig. 5C) and Campbell Island (Fig. S5) with highly significant relationships to a sector in the southwest Pacific (50˚-60˚S, 150˚-170˚E; Table 1), supporting our earlier observation of the thermal coupling between atmospheric and ocean temperatures but extending across the broader region. Although we find no evidence for a sustained shift in airflow direction that parallels the observed trend in subantarctic temperatures (Fig. S8) we do observe a marked increase in wind strength across the late twentieth century, with a long-term intensification (with high variability) of winds that closely parallels air temperatures over Macquarie Island (Fig. 5D); the original AAE data is plotted for completeness but given uncertainties over the reliability of historic observations (Jakob, 2010) a direct comparison is not possible. This trend towards stronger winds is accompanied by an increase in sunshine hours over Macquarie Island (Fig. S9), consistent with reduced cloud cover, but any associated increase in sensible heat flux appears to be substantially modulated by increased airflow over cooler surface waters in the southwest Pacific (Thompson et al., 2011). Our results, therefore, are in line with the observed (post-1979) spring-summer trend towards windier conditions in the southwest Pacific (Fig. S1).

Whilst some studies have suggested a dynamical atmospheric circulation response to ozone layer depletion over the Southern Hemisphere mid-latitudes since the 1990s (Thompson et al., 2011), the reconstructed 2.4 and 3-year periodicities suggest a tropical teleconnection with the southwest Pacific (Kestin et al., 1998; Adamson et al., 1988). Using the HadISST (Rayner et al., 2003) and ERA Interim (Dee et al., 2011) datasets, a significant inverse correlation is observed between subantarctic and central-eastern low latitude Pacific temperatures and zonal wind stress, with a relatively warm (cool) eastern equator associated with weaker (stronger) mid-latitude westerly winds and cooler (warmer) SSTs in the southwest Pacific (Figs. 5A-C and S5). Comparison to different measures of tropical Pacific SSTs and atmospheric circulation indicate the most significant relationship with subantarctic spring-summer temperatures is the Nino 3 region (correlation -0.592, $p$ < 0.001) (Table 1).

To elucidate the mechanism by which changes in the tropical Pacific may be projected onto the high latitudes, we explored the relationship between Nino 3 temperatures and Southern Hemisphere atmospheric circulation using data from ERA Interim (Dee et al., 2011) (Fig. 6). We observe what appears to be a Rossby wave train similar to the PSA climate mode of variability during the austral spring-summer (Ding et al., 2012; Mo and

Higgins, 1998; Trenberth et al., 1998). We find that post-1979, warmer temperatures in the Nino 3 region leads to deep convection and upper-level divergence flow (at 300 hPa) (Fogt et al., 2012; Ding et al., 2012; Trenberth et al., 1998) (Fig. S16), apparently forcing an atmospheric Rossby wave train southeast into the extratropics manifested as cyclonic anomalies south of New Zealand – consistent with the relationship observed with Macquarie Island temperatures (Fig. 5) – that extend across the Pacific as anticyclonic anomalies in the Amundsen-Bellingshausen seas and cyclonic anomalies off the east coast of South America (Ciasto and Thompson, 2008; Mo and Higgins, 1998). Lead-lag analysis demonstrates the atmospheric signal propagates over southern New Zealand during the late austral winter and reaches the Amundsen-Bellingshausen seas by the summer (Fig. S17). Our results support previous studies that find the PSA signal precedes peak temperatures by approximately one season and abruptly weakens during the austral summer (Schneider et al., 2012) (Figs. S4 and S17).

With the above tropospheric pressure changes (Fig. 6) we suggest warmer Nino 3 temperatures are associated with stronger westerly airflow over the southwest Pacific subantarctic islands and west Antarctic coast, accompanied by enhanced southerly airflow across the Antarctic Peninsula that extends into the south Atlantic. Hovmöller plots show an alternating pattern of warm-cold surface temperatures between the southwest Pacific and Amundsen-Bellingshausen seas using both the HadISST (Rayner et al., 2003) and Reynolds v2 SST (Smith and Reynolds, 2005) datasets (Fig. S14), consistent with atmospheric Rossby wave propagation and regional ocean surface responses. Running 30-year correlations between the *Dracophyllum* series and measures of westerly airflow, however, suggests no relationship with a hemispheric-wide reconstruction of SAM that extends back to CE 1884 (Fig. 7A) (Marshall, 2003; Visbeck, 2009). Regional monthly changes in the structure of SAM are now recognized and allow sector-specific analysis (Fogt et al., 2012; Ding et al., 2012; Visbeck, 2009). Here we identify a significant inverse correlation to the Australasian region for the austral winter and spring during the post 1940s period ($p < 0.05$; Fig. 7C), while the Southern Hemisphere-wide and regional South American and African SAM reconstructions do not appear to be significant for any period across the twentieth century (Figs. 7B and C). Previous work has demonstrated that the PSA is an important contributor to the zonal asymmetry in SAM (Fogt et al., 2012; Ding et al., 2012), suggesting the tropics are indeed imposing a signal on mid-latitude westerly airflow in the southwest Pacific. However, in contrast to earlier studies that have postulated anthropogenic forcing may have changed the structure of SAM to be more zonal (Fogt et al., 2012), our results imply the tropics have introduced an asymmetry to the Australasian sector of SAM in the modern record, or this has at least become more common during the second half of the twentieth century.

To investigate whether the changes in the southwest Pacific subantarctic region are representative of a larger part of the Southern Hemisphere we analysed simulations with the three-dimensional Earth-system model of intermediate complexity LOVECLIM1.3 for CE 1850– 2009, driven by natural (solar and volcanic) and anthropogenic (greenhouse gases, sulphate aerosols, land use) forcings (Zunz and Goosse, 2015) (Fig. 8). For the 1850–2009 simulation, the model was forced to follow the observations of surface temperature. We examined the changes in zonal wind stress between selected decades across the twentieth century, including 1910-1919 (overlapping the original AAE period) (Fig. 8). Over the past century, we find increasingly stronger westerly winds across the Southern Ocean with a marked intensification in the southwest Pacific and Antarctic

Peninsula during the most recent decades with more easterly airflow over the Ross Sea (Fig. 8C), trends also
observed in estimates derived from the ERA Interim dataset (Fig. 8D) (Dee et al., 2011), and consistent with the
observational record from Macquarie Island (Figs. 5D).

## 3.5 Pacific-wide changes

Although there appears to have been a long-term strengthening of westerly winds across key sectors of the mid-
latitudes, the Macquarie Island record suggests this has also been accompanied by increasing variability (Fig.
5D). To explore whether this is manifested across the wider Pacific we compared our 140-year temperature
reconstruction to key datasets (Fig. 9). Parallel changes in SST magnitude and trend in the southwest Pacific
using both the LOVECLIM model output and HadISST (Rayner et al., 2003) is consistent with our
reconstruction of subantarctic island temperatures (Figs. 3, 8 and 9). Intriguingly, the inferred increasing
westerly winds and warming Southern Ocean in the southwest Pacific have been accompanied by a regional
order of magnitude decline in marine vertebrate populations (Morrison et al., 2015), suggesting the increased
inter-annual temperature variability may have played a role, and will form a focus for future work. Importantly,
we find a comparable increase in temperature and variance in the Nino 3 region, supporting our contention that
the tropics are a major driver of variability across the subantarctic Pacific and implying similar variability may
be expressed across other sectors of the Southern Ocean, albeit lagged by 1-3 months (Fig. S17). To test this we
utilise snow core accumulation records from coastal West Antarctica, a region identified as sensitive to
atmospheric pressure anomalies associated with the PSA (Thomas et al., 2008) (Fig. 6). Previous studies have
reported a mid to late twentieth century increase in precipitation associated with a deepening of the Amundsen
Sea Low (ASL) (Thomas et al., 2008; Thomas et al., 2015), where strong northerlies advect warm south Pacific
air masses over the continent, resulting in orographic-driven precipitation over the southern Antarctic Peninsula
(Gomez ice core; Fig. 9F) and the West Antarctic coastal sites Bryan Coast and Ferrigno (Fig. S18). Importantly
the observed twentieth century increase appears to be confined to the Antarctic Peninsula and West Antarctic
coast, with the magnitude decreasing from east (Gomez) to west (Ferrigno); in marked contrast, the observed
increase is not recorded in the continental interior (Thomas et al., 2015). Whilst the ASL is generally considered
quasi-stationary because of the large number of low pressure systems in this sector of the circumpolar trough
(Hosking et al., 2013), the snow core derived increases in precipitation are accompanied by an increase in 30-
year running mean of the standard deviation, suggesting increased variability in the ASL region that is unusual
in the context of the past 300 years, with the Gomez site most sensitive to changes in synoptic conditions.

Whilst we cannot preclude that the climate teleconnections may have been different prior to the 1940s, the
parallel changes in variance observed across the Pacific suggests this is not likely (Fig. 9). This interpretation is
supported by the recently reported stepped increase in spring-summer rainfall over the south Atlantic during the
1940s, a shift apparently unprecedented over at least the last 6000 years, and interpreted to be a consequence of
highly seasonal changes in atmospheric pressure over the Amundsen-Bellingshausen seas (Turney et al., 2016a).
Although analysis of the most recent decade suggests a weakening of the PSA (Trenberth et al., 2014), the
observed persistently high spring-summer Pacific variance and increase in Atlantic precipitation (Turney et al.,
2016a) suggests that Rossby wave penetration of the high latitudes remains substantial when placed in the
context of the last 140 years (Fig. 9).
**5 Conclusions**
Our study adds to a growing body of literature that increasing and variable tropical temperatures are a major
driver of spring-summer Southern Hemisphere atmospheric circulation changes (Jones et al., 2016; Steig et al.,
2009; Wang and Dommenget, 2015; Schneider et al., 2012; Ciasto and Thompson, 2008). Our findings,
however, provide a long-term perspective that suggests modern observed high interannual variability was
established across the 1940s, and that the influence of contemporary equatorial Pacific temperatures may now
be a permanent feature across the mid to high latitudes. Further work is now required to extend key records and
explore climate variability back through the Holocene (Cobb et al., 2013). This study emphasises the
considerable value of tree ring and historical data for extending satellite observations of the Southern Ocean
beyond 1979 (Goosse and Zunz, 2014), and the use of ocean and climate models to interpret trends in rapidly
changing terrestrial and marine environments, including sea ice (Turner et al., 2015; Hobbs et al., 2016). Our
results offer the potential to improve forecasts across the extratropical region (Trenberth et al., 1998) and have
implications for the interpretation of proxy data from locations with non-stationary relationships to modes of
Southern Hemisphere atmospheric circulation.
**Author Contribution**
CT and CF conceived the research; CT, JP, EvS, ZT, SR, PV, VZ, HG, K.-J.W. designed the methods and
performed the analysis; CT wrote the paper with input from all authors.
**Competing Interests**
The authors declare that they have no conflict of interest.
**Acknowledgements**
A large thanks to the captain and crew of the *MV Akademik Shokalskiy*, and Henk Haazen and Kali Kahn on the
*Tiama* for all their help in the field. Thanks also to Lisa Alexander (CCRC) for the analysis of meteorological
datasets and to Jean-Baptiste Sallee who kindly provided the location of the main fronts of the ACC (Fig. 1).
This work was supported by the Australasian Antarctic Expedition 2013-2014, the Australian Research Council
(FL100100195, FT120100004, DE130101336 and DP130104156) and the University of New South Wales.
Research on the New Zealand subantarctic Campbell Island was undertaken under Department of Conservation
National Authorisation Numbers 37687-FAU and 39761-RES. Kathy Allen and three anonymous reviewers
kindly provided valuable and constructive criticism which improved an earlier version of this manuscript.

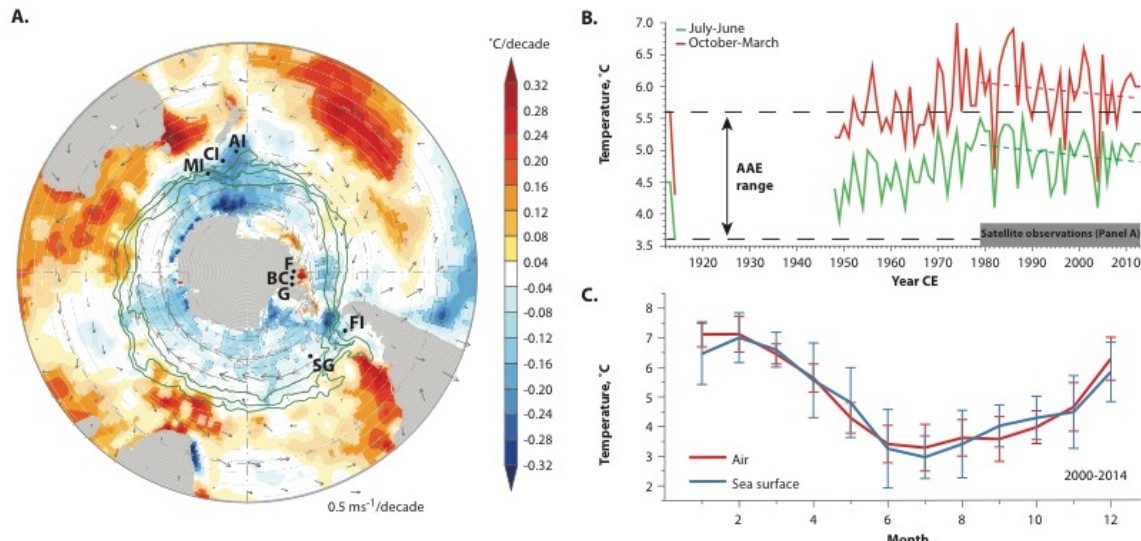


**Figure 1. Ocean-atmosphere coupling in the Southern Hemisphere. A.** Significant ($p < 0.05$) austral summer (December-February) sea surface temperature (SST ˚C/decade; shading) and 925-hPa winds (vectors) trends since 1979. Temperatures based on SSTs from the HadISST dataset (Rayner et al., 2003); winds from ERA Interim (Dee et al., 2011). Key sites discussed in text are shown: Macquarie Island (MI), Campbell Island (CI), Antipodes Island (AI), Ferrigno (F), Bryan Coast (BC), Gomez (G) (Thomas et al., 2008; Thomas et al., 2015), Falkland Islands (FI) and South Georgia (SG) (Turney et al., 2016a). Overlaid in green are the three main fronts of the Antarctic Circumpolar Current (Sallée et al., 2012). **B.** Annual (July-June) and spring-summer (October-March) air temperatures at Macquarie Island. Dashed lines denote range of the Australasian Antarctic Expedition temperatures (AAE; CE 1912-1915). Period of satellite observations (Panel A.) shown by grey bar; dashed coloured lines denote trend in temperatures across the satellite period. **C.** Monthly Macquarie Island air (red line) and sea surface temperatures (blue line) (with 1σ) demonstrating tight coupling between atmospheric temperature and SSTs (2000-2014).

545

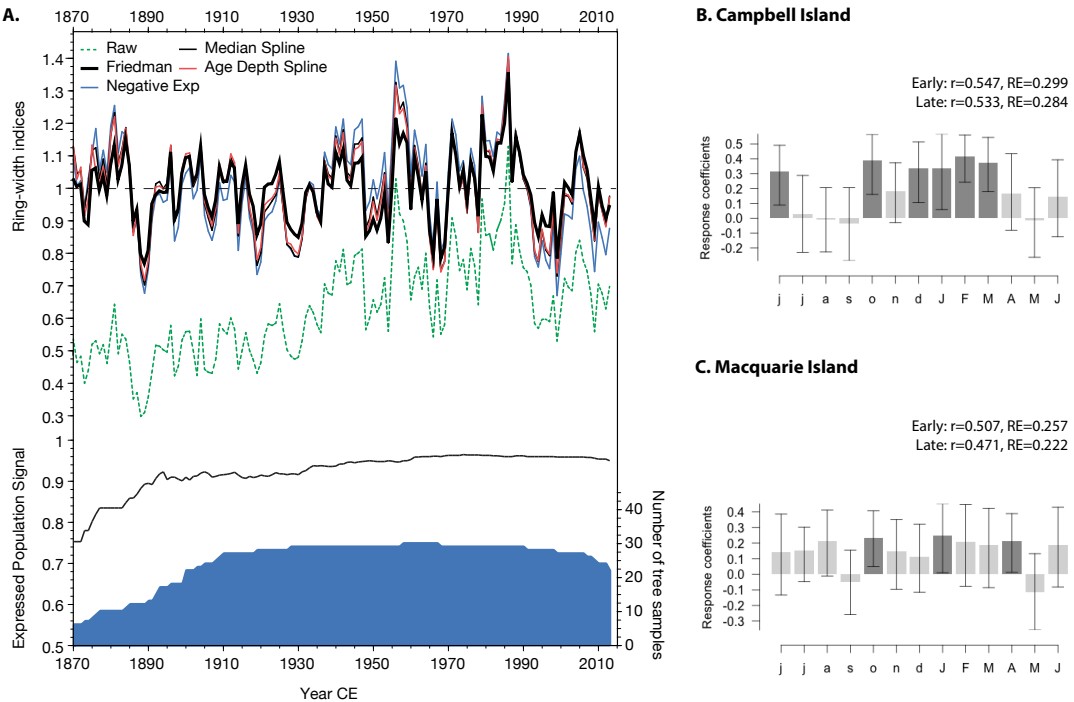

546

**Figure 2. Developing a temperature-sensitive tree-ring record from the subantarctic Pacific. A.** *Dracophyllum* raw tree-ring chronology (green line) with different standardization outputs (various coloured lines), Expressed Population Signal (EPS; thick red line) and sample size of trees (blue area). Bootstrap correlation function of the *Dracophyllum* tree-ring chronology to instrumental records of monthly temperatures from Campbell Island (**B.**) and Macquarie Island (**C.**) with error statistics for early (CE 1949-1980) and late (1981-2012) calibration periods. Darker bars indicate months with statistically significant correlations ($p < 0.05$).

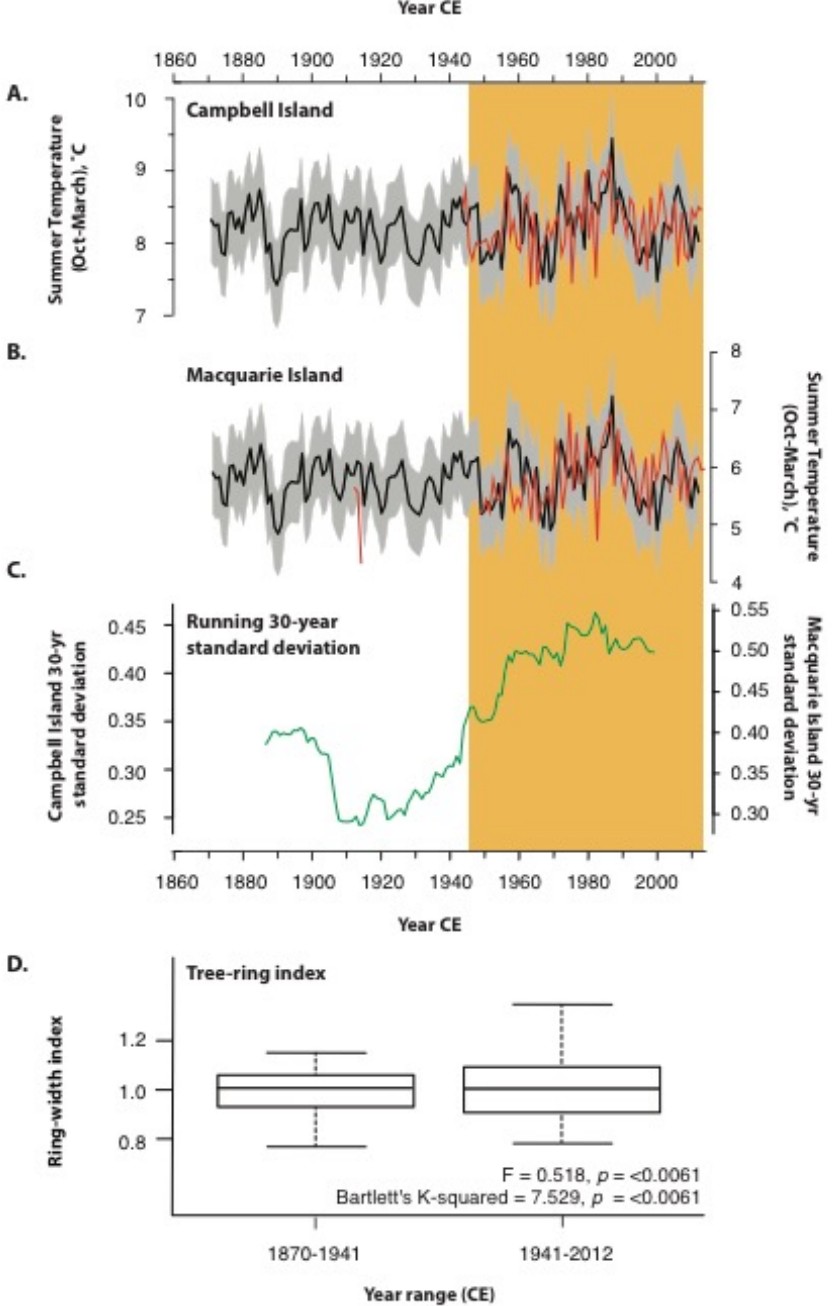

553

**Figure 3. 140-year temperature variability in the subantarctic Pacific.** Campbell Island (**A.**) and Macquarie Island (**B.**) observed (red lines) and *Dracophyllum* reconstructed (black) growing season temperatures (October-March) with 90% quantile limits (grey envelope) compared against running 30-year mean standard deviation of the reconstructed temperature series (**C.**). **D.** Box and whisker plots of the ring width indices with summary statistics indicating a significant difference in variance between the periods CE 1870-1941 and 1941-2012. Orange column defines significant post-1940s temperature variability in the record.


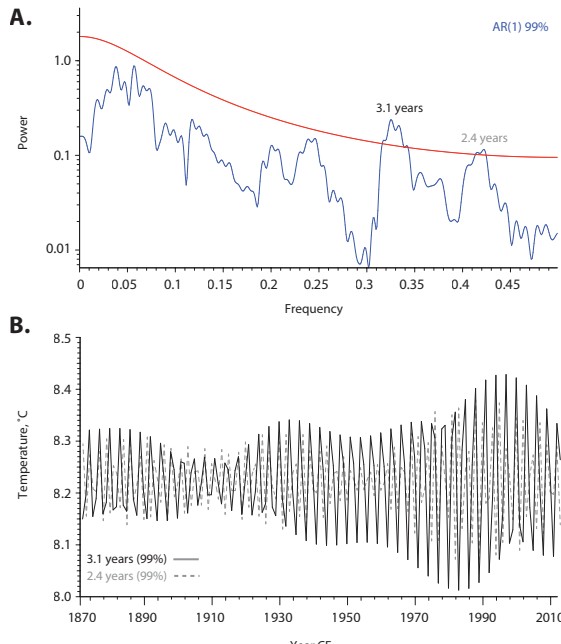


**Figure 4. Tropical variability in the subantarctic temperature record.** Changing amplitude of reconstructed summer
temperatures for Macquarie Island. Multi-taper method (MTM) (**A.**) and extracted climate periodicities exceeding 99%
significance (**B.**) observed in the *Dracophyllum*-derived growing season temperature reconstruction for Macquarie Island
since CE 1870.

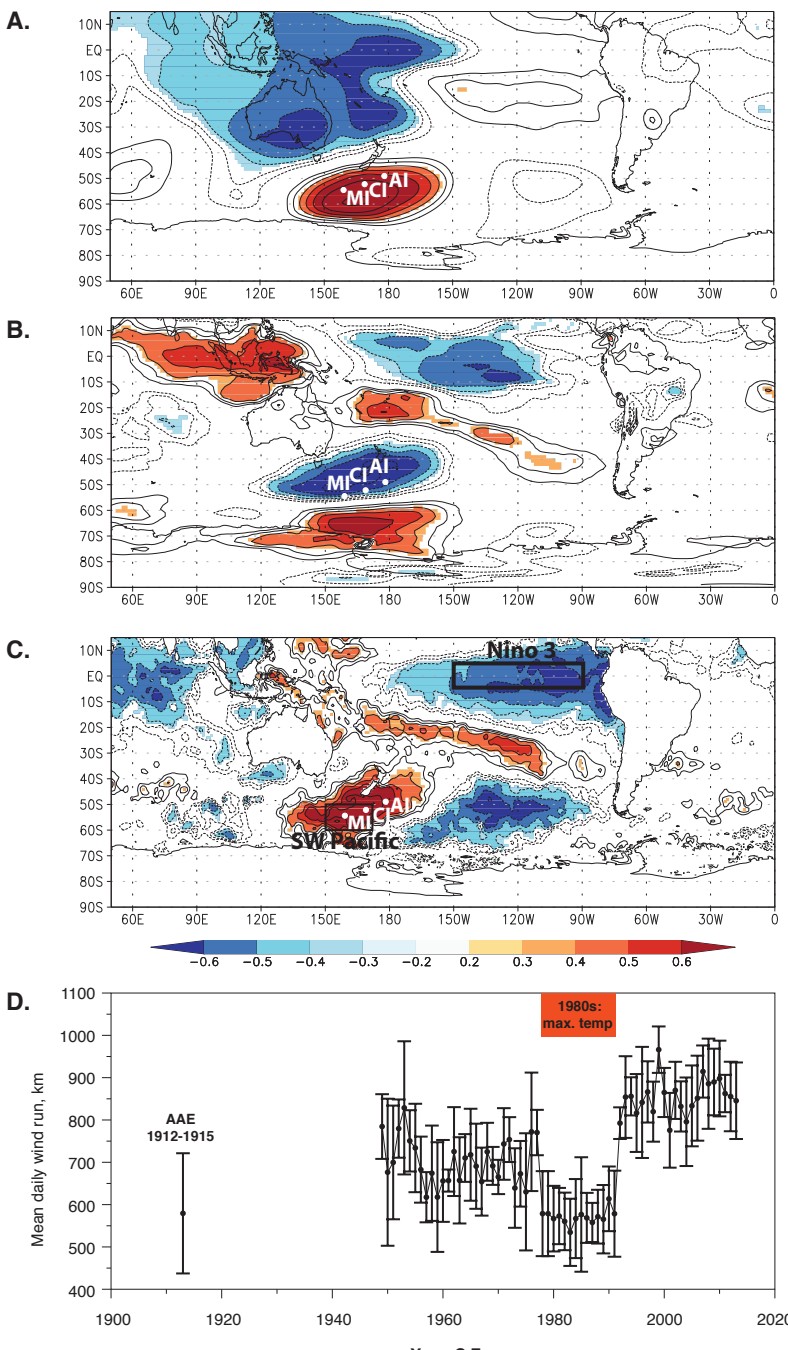


**Figure 5. Climate controls on temperature over Macquarie Island.** Spatial correlations between detrended and deseasonalised Macquarie Island mean monthly atmospheric temperatures (October-March) and 850 hPa height (**A.**), zonal wind stress using ERA Interim[31] (**B.**) and sea surface temperature (HadISST; **C.**) (Rayner et al., 2003) for the period 1979-2013 ($p_{field} < 0.05$). Note: Campbell Island (CI) and the Antipodes Islands (AI) fall within the region of greatest correlation to SSTs in the southwest Pacific. The southwest Pacific (SW Pacific; 50-60˚S, 150-170˚E) and Nino 3 regions also shown. For comparison, mean seasonal (October-March) daily wind run (kilometres) for the meteorological station at Macquarie Island (source: Bureau of Meteorology) with comparison to average from the Australasian Antarctic Expedition (1912-1915) with 1σ uncertainty (**D.**). Note, the period of decreased wind speed across the 1980s coincides with maximum air temperatures over Macquarie Island (see Fig. 3).

576

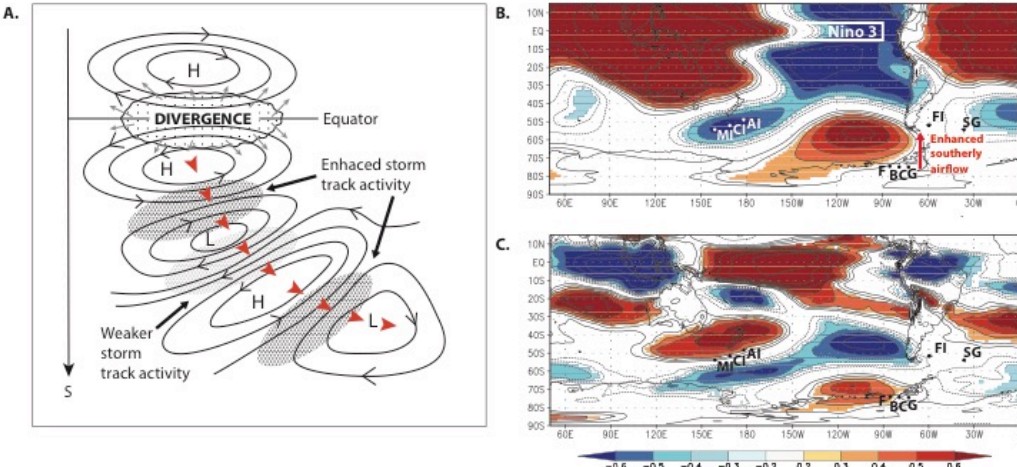

**Figure 6. Rossby wave propagation from the tropical Pacific during the austral spring-summer.** Low to high latitude atmospheric teleconnections during the austral spring and summer (October-March). Schematic showing extratropical Pacific-South America (PSA) Rossby wave train (red arrows) associated with low and high pressure systems generated by anomalous equatorial upper-level divergence flow (Trenberth et al., 1998); enhanced southerly airflow across the West Antarctic coastline extends into the South Atlantic during anomalously high temperatures in the Nino 3 region (**A.**). Spatial correlations between detrended and deseasonalised Nino 3 sea surface temperature (Rayner et al., 2003) (October-March) and 850 hPa height (**B.**) and zonal wind stress (**C.**) using ERA Interim (Dee et al., 2011) for the period 1979-2015. Location of key sites are shown. Significance $p_{field} < 0.05$.

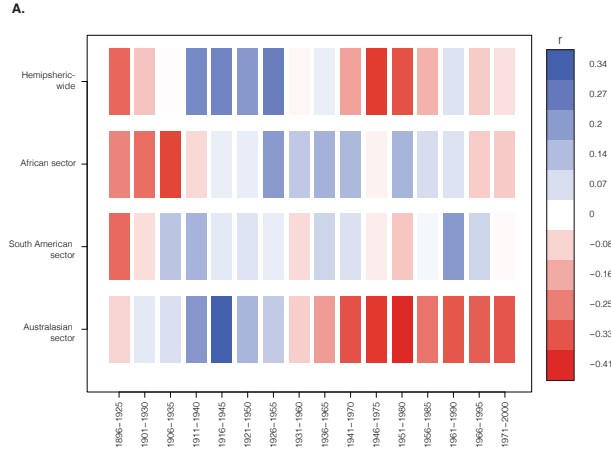

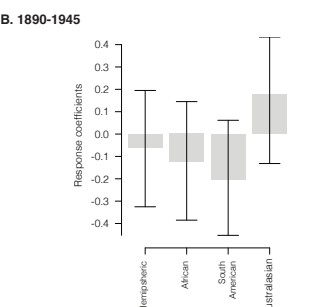

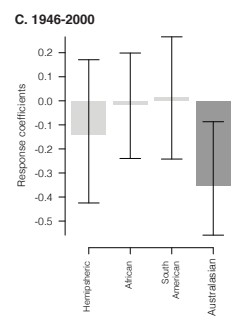

586

**Figure 7: Changing Southern Annular Mode (SAM) relationships through the twentieth century.** Running 30-year correlations (**A.**) and bootstrap correlations (**B.** and **C.**) between hemispheric-wide and sector-specific SAM reconstructions (July-October) (Visbeck, 2009) and the *Dracophyllum* series. Bootstrap correlation periods obtained by halving the SAM dataset spanning 1890 to 2000. The dark bar indicates only the Australasian SAM has a statistically significant correlation to the temperature-sensitive tree-ring series during the post-1940s period for the austral winter and early spring ($p < 0.05$).

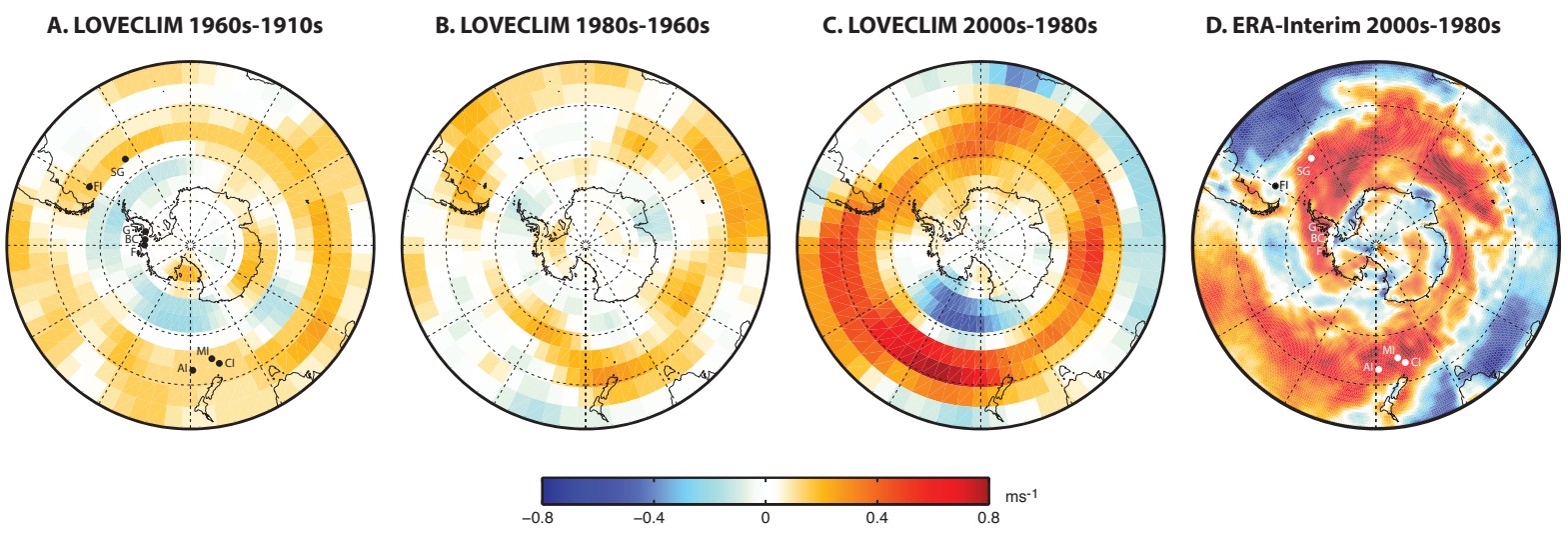

592

**Figure 8. Modelled changes in Southern Hemisphere westerly airflow over the last century.** Differences in zonal October-March wind speed (ms[-1]) at 800 hPa across the Southern Ocean derived from LOVECLIM1.3 (Zunz and Goosse, 2015) (Panels A.-C.) and ERA-Interim (Dee et al., 2011) (Panel D.). Location of key sites discussed in text are also shown.

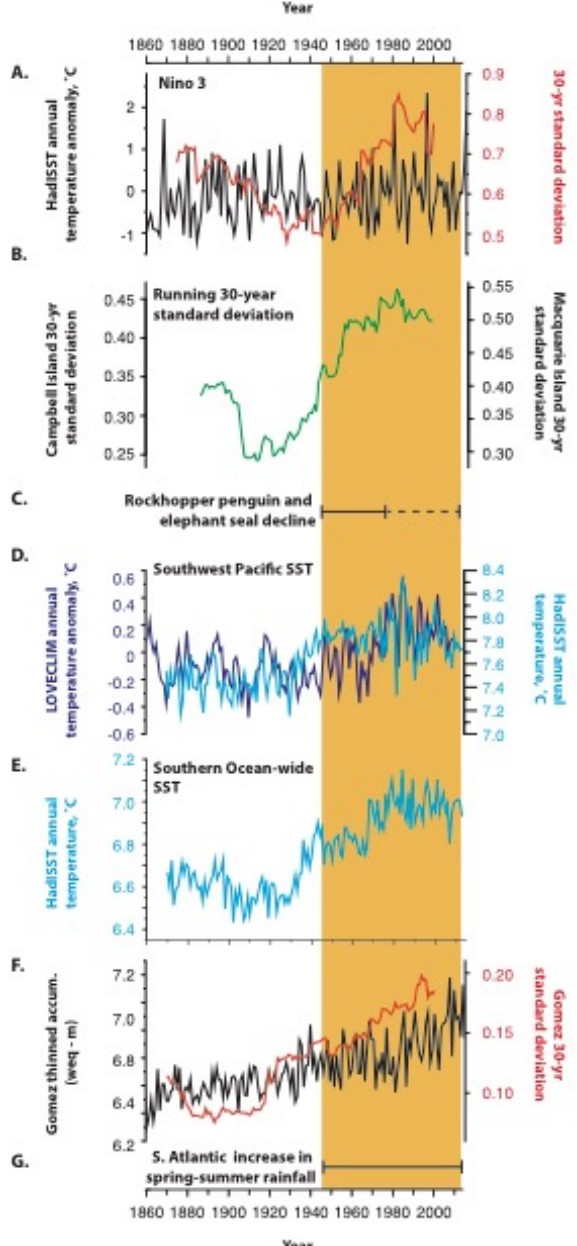

595

**Figure 9. Equatorial and south Pacific temperature and marine population trends since CE 1860.** Nino 3 temperature (July-June) with running 30-year mean standard deviation of the HadISST temperature series (Rayner et al., 2003) (**A.**) compared against Campbell Island and Macquarie Island running 30-year mean standard deviation of the reconstructed temperature series (**B.**). Orange column denotes twentieth century temperature variability that exceeds any other period in the record. Onset (solid line) and continuing (dashed) period of declining rockhopper penguin and elephant seal populations in the southwest Pacific (Morrison et al., 2015; Weimerskirch et al., 2003) (**C.**) shown for comparison. In addition, mean annual temperature (°C, July-June) sea surface temperatures (HadISST and LOVECLIM model output) for the Campbell-Macquarie islands region (**D.**) and wider Southern Ocean (**E.**). Note the coincident increase in west Antarctic coast (Gomez) (annual and 30-year mean standard deviation) (Thomas et al., 2015; Thomas et al., 2008) (**G.**) and south Atlantic (Falkland Islands and South Georgia) precipitation (Turney et al., 2016a) (**F.**).

|  | SW Pacific | Nino 4 | Nino 3.4 | Nino 3 | Nino 1+2 | SOI |
|---|---|---|---|---|---|---|
| *Macquarie Is* | | | | | | |
| July-June | **0.813‡** | -0.392* | -0.512† | **-0.563‡** | **-0.558‡** | 0.470† |
| October-March | **0.835‡** | -0.396* | -0.523† | **-0.596‡** | **-0.614‡** | 0.475† |
| *Campbell Is* | | | | | | |
| July-June | **0.754‡** | -0.412* | -0.514† | **-0.546‡** | -0.531† | 0.458† |
| October-March | **0.782‡** | -0.409* | -0.534† | **-0.592‡** | **-0.582‡** | 0.473† |

**Table 1**: Correlations and significance of relationship between subantarctic island air temperatures and measures of regional (50-60˚S, 150-170˚E) and equatorial Pacific sea surface temperature (SST) and atmospheric circulation. Regional and Nino temperature anomalies as calculated from HadISST (Rayner et al., 2003); the Southern Oscillation Index (SOI) as reported by Ropelewski and Jones (1987). Deseasonalised and detrended correlations derived for the period CE 1979 to 2014. Significance indicated as follows: *$p<0.05$, † $p<0.01$, and ‡ $p<0.001$.

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
