# Peer review of "Tropical forcing of increased Southern Ocean climate 1 variability revealed by a 140-year subantarctic temperature 2 reconstruction 3"

_Climate of the Past, 2016_

## Referee Comment (RC1) · Anonymous Referee #1 · 16 Dec 2016

*Review of Tropical forcing of increased Southern Ocean climate variability revealed by a 140-year subantarctic temperature reconstruction by Turney et al.*

**General comments**

This paper used a new dendrochronology from trees in sub-Antarctic islands Campbell and Macquarie to build a temperature reconstruction back to 1870. The paper addresses a need for more information about past temperature variability in the high southern latitudes, and uses a number of approaches (including models, reanalysis, weather observations and ecological data) to support the findings made by the reconstruction. The article is well written and prepared, and generally makes a good effort to enhance the reproducibility of the study by describing the tools and programs used.

However, I am concerned that the authors have not considered the scarcity and poor quality of observational data in the high southern latitudes in some aspects of their analysis. This issue may confound some of the conclusions drawn, and affect the independence of the different lines of evidence they use to make their arguments. Overall I believe it should be published subject to major revisions.

**Specific comments**

Line 98: You say that Macquarie Island is highly sensitive to Rossby wave propagation, but the correlation values don't seem that high in Figure S4 (~0.3–0.4). Can you expand on this claim, or soften it?

Section 2.1: This section is a bit unclear. What quality control has been done to the AAE data? Where are they available? You also mention comparing the satellite SSTs to the AAE reports, but was SST recorded during 1912–1915? Is this the Buckles Bay measurements, or are they the Bureau of Meteorology records? A version of Table S1 here might make this section more clear.

Section 2.2. This section (and Tables S2 and S3) confused me. What are you trying to show? That the interannual variability of the AAE data is within the modern range? That there is a temperature trend? What are the statistical results relative to?

Figure 3: Do you have a possible explanation for the disagreement between the obs and the reconstruction in the 1950s?

Section 3.2. To me this section makes more sense at the start of section 2.3, as it does not add much here.

Section 3.3. From where does HadISST get its data? Given the data scarcity in these high latitudes, it would be worth confirming that HadISST does not have the same data as Loewe, or has not approximated the Bureau's air temperatures as SSTs, given the high level of agreement between the two.

Section 3.3 as well. Chelton and Raisin (2016) mention that there is an increase in standard deviation in HadISST from 1949, at least over the North Pacific, due to artifacts in the dataset. It would be worth acknowledging this or even better, arguing that your dendrochronology results show a real variance increase that is independent of this artifact in the observational data.

Section 3.4. This is a nice additional independent line of evidence, I look forward to reading the future research.

Line 376–380. Historical wind data are notoriously rubbish (e.g. Jones et al. 1997; Jakob, 2010). I'm suspicious of drawing any conclusion on AAE wind data unless you have good metadata about where exactly the wine vane was located.

Jakob D. 2010. Challenges in developing a high-quality surface wind-speed data-set for Australia. *Australian Meteorological and Oceanographic Journal* 60: 227–236.

Gallego D, Garcia-Herrera R, Calvo N, Ribera P. 2007. A new meteorological record for Cadiz (Spain) 1806–1852: Implications for climatic reconstructions. *Journal of Geophysical Research* 112, DOI:10.1029/2007jd008517.

Lines 444–445: You say that "The close similarity between the LOVECLIM output and HadISST argues against any bias in the latter dataset for this region", however you have forced LOVECLIM with HadCRUT3. As I understand it, HadCRUT3 gets its SST data from HadSST2, which in turn gets its data from ICOADS. HadISST also uses COADS data (the precursor to ICOADS) to increase data coverage, although I'm not sure exactly where. You need to check the data sources for the region for both datasets to claim that these results are independent.

**Technical corrections**

Line 10: I would reverse this sentence to improve readability: "SAM and ENSO play a dominant role in this as modes of large scale variability".

Line 129: Macquarie Island were, not was.

Lines 197 and 198: I think you mean Figure S11 and S12.

Line 263: I'd put a comma after limiting, the sentence is a bit confusing otherwise.

Line 289: add a 'to' after the word limited

Line 359: data are

Figure 1, and Figures S1–S3: What is the source of the data fields you are plotting? I'd also add ºC to the colour bar just for clarity.

Figure 1: South Georgia is labeled as SG in the image, but GI in the caption.

Figure 2: Consider using dotted/dashed lines for colour-blind readers.

Figure 3: I would make the orange block lighter to make the graph easier to look at.

Figure 4: Consider changing the colours here to grey and black for colour-blind readers.

Figure S4: Which areas are significant? Those that are shaded? Please clarify. You could even add the significance threshold correlation value.

Tables S5 and S6 could be swapped so they are in the order in which they are mentioned. This goes for the figures and supplementary figures as well. At times I got a little lost because the Figure references were not in order (e.g. Figures 8 and 9 mentioned before Figures 5–7).

---

## Referee Comment (RC2) · Anonymous Referee #2 · 24 Dec 2016

General comments This is a very interesting and carefully written manuscript that sheds significant light on pre-instrumental climate variability over the southern oceans. The evidence for an increase in climate variability is compelling, as are the links to tropical/ENSO variability and Rossby wave propagation. There is quite some overlap between sections 2 and 3 and there is some repetition. Can you rearrange the text to keep the results confined to section 3?

Specific comments Lines 59-60: Could also add Jones et al. (2016) here. Line 84: I wouldn't use the word "competing" here. Both flows act to reduce the north-south temperature gradient, hence are really working in the same sense.

Technical corrections Line 211: Define NCEP. McGlone et al (2010) reference is

missing an author.

Please also note the supplement to this comment:
http://www.clim-past-discuss.net/cp-2016-114/cp-2016-114-RC2-supplement.pdf
——————————————————

[Figure]

**Supplement:**

Review of manuscript *Tropical forcing of increased Southern Ocean climate variability revealed by a 140-year subantarctic temperature reconstruction*, by Turney et al., submitted to *Climates of the Past*, November 2016.

**Overall recommendation:** Accept, pending minor revisions.

**Remarks to the author**

*General comments*

This is a very interesting and carefully written manuscript that sheds significant light on pre-instrumental climate variability over the southern oceans. The evidence for an increase in climate variability is compelling, as are the links to tropical/ENSO variability and Rossby wave propagation.

There is quite some overlap between sections 2 and 3 and there is some repetition. Can you rearrange the text to keep the results confined to section 3?

*Specific comments*

**Lines 59-60:** Could also add Jones et al. (2016) here.

**Line 84:** I wouldn't use the word "competing" here. Both flows act to reduce the north-south temperature gradient, hence are really working in the same sense.

*Technical corrections*

**Line 211:** Define NCEP.

McGlone et al (2010) reference is missing an author.

---

## Referee Comment (RC3) · Anonymous Referee #3 · 29 Dec 2016

Review of Turney et al., Tropical forcing of increased Southern Ocean climate variability revealed by a 140-year subantarctic temperature reconstruction

I have reviewed a previous version of this paper before for another journal. I am pleased that the authors have addressed many of my previous concerns, including taking up my suggestion to link temperatures at the subantarctic Islands to local sea surface temperatures, as well as to tropical climate variability.

The temperature reconstruction is valuable and good work, adding to understanding of climate vaiability in this data sparse region, and deserves to be published, as does the discussion of the increased variability and influencing factors on climate variability in this region. However the structure of the paper needs to be clearer, there is much

analysis in the discussion, and the phrasing and interpretation of results in some places needs to be clearer and reflect what can be drawn from the results in all places. Also, the figures are not always numbered in the order of discussion in the text, which also hinders following of the discussion. Also, the relative influence of local vs tropical SSTs should be considered more fully.

I therefore recommend publication subject to major revisions. I apologise that this review is late.

1. Make sure that all figures, both in the main text and supplementary information, are numbered in order of discussion in the text, this is not done for all figures.

2. Section 3.1. Modern climate changes. Figure 1c shows well the coupling between atmospheric and oceanic temperatures at Macquarie Island over the annual cycle. Similarly Figure 5C shows, as well as the influence of SSTs in the Nino 3 region, strong correlations with local SSTs. Given this close coupling, the influence of this strong coupling on interannual temperature variability on Macquarie Island (MI) and Campbell Island (CI) needs to be explored further in the paper, and consideration of the relative proportions of temperature variance explained by local vs remote SSTs is needed. You should therefore include the timeseries of MODIS and HadISST temperatures for gridboxes/pixels closest to the islands in Figure 1. I would like to see this for both MI and CI. You could add the CI plots to the supplementary info, although additional panels in Figure 1 would also work. In Table 1, please include correlations with local SSTs (Modis and HadISST). I realise the MODIS correlations will be for a shorter period, but I think this analysis would still add to the paper. This analysis will allow quantification of the relative influence of local versus remote SSTs. I realise that SSTs in these two regions may not be completely independent, so perhaps also include a column correlating NINO 1+2/3 and SSTs local to MI/CI too.

I would also like to see (perhaps as a panel in Figure 9), the tree ring reconstructions with HadISST SSTs for the MI/CI region.

3. The penguin and seal analysis is not convincing and I recommend either removing this from the paper, or toning this section down. Although I am not an expert in links between penguin populations and climate, I am not convinced that the coincidence of the decline with an increase in climate variability does prove a causal linkage.

Additionally, from reading one of the sources cited (Morrison et al. 2015), these authors state that although there has been an overall long-term decline, Rockhopper Penguin numbers on Campbell Island have recovered/stabilised over the period 1996-2012 (a fact not explicitly mentioned in the text, rather hinted at by stating that 94% of the decline had occurred by the mid 1980s). Morrison et al. state 'the recent decline occurred during the period 1984–1996 and was followed by overall population growth 1996–2012, concurrent with lower SSTs and an increased abundance of a key prey species'. This would seem to fit with the SST changes discussed in Point 1 above, rather than to temperature variability.

4. Clarity of phrasing.

- First sentence of the paper. Figures 1 and S1 do not show ice sheet dynamics.

- Line 68. 'Late twentieth century Southern Ocean climate' – I assume you mean climate over the Southern Ocean region, not the climate of the ocean itself.

- Line 63, make it clear what 'This' is referring to.

- Page 2 line 75. 'ENSO is associated with spatially different temperature and wind trends across mid to high latitudes (Figure S3). Figure S3 shows the regression of Nino 3 and 3.4 on SST and wind - therefore you mean the signal/influence, not the trend.

- Section 2.2. line 150, comparison of AAE temperatures to those on the islands. Firstly, make it clear in the text that it's temperature measurements. You state that 'this analysis illustrates the trend through time rather than interpreting specific comparisons'. I do not understand what you mean by this, please rephrase.

- Line 291. You state 'a long-term trend towards increasing temperatures from the 1960s that reached a maximum during the late 1980's. I wouldn't call a 20 year trend a long-term trend, and temperatures reach a maximum in the late 1980s, not the trend. It would also help the reader if you mark 1912-15 on Figure 3. You then state that the late 1980s were warmer compared to 1912-1915, indicating some kind of linear behaviour – but what Figure 3 shows to me is lots of variability, and there are periods since 1912-15 that are cooler, and others that are warmer. Please rephrase this more carefully and clearly.

- Line 481, 'contemporary equatorial Pacific temperatures may now be a permanent feature across the mid to high latitudes' – do you mean the influence of contemporary equatorial Pacific temperatures. This sentence currently doesn't make sense.

- Caption of Fig S7, you state that there is no sustained change in wind direction since the expedition. I do not think that this can be concluded from this plot. Please rephrase (our just delete). See point 6 below.

Discussion

5. An overall comment to the discussion section is that I suggest having sub-headings in this section, to guide the reader, as well as addressing the issue of order of figure numbering, as currently this section is a little difficult to follow. You may wish to consider instead of having separate results and discussion sections, restricting to having an overall 'Results and Discussion' section (as there is a lot of analysis in the results section), but with clear subheadings that lead the reader through the analysis.

6. Line 377. You state there's no long term trend in solar radiation – it appears from Figure S8 that there has been a shift towards greater sunshine hours since the mid-1960s. This needs exploring/explaining. Also in this paragraph, you state that there has been 'a long term intensification' of winds. Please remove the phrase 'long-term intensification'. What can be concluded from Figure 5D is that winds since 1950 are stronger than those during 1912-1915, and that the 1981-2010 winds are stronger

than the 1951-1980. This could be evidence of a long-term intensification, but given the strong interannual variability in this region, there does need to be the caveat that the 1912-1915 winds are a snapshot of winds in a region with strong variability.

Also in this paragraph, Line 380/discussion of Figure S1. It needs to be shown on Figure S1 which trends are significant, perhaps marking these with a different colour arrow. Also, marking of MI and CI on this plot would be useful.

7. Line 394. You state that you 'observe a Rossby wave train', and online 396, that 'We find that post-1979 warmer temperatures in the Nino 3 region leads to deep convection. . .. Forcing an atmospheric wave train'. The discussion in this paragraph needs to be rephrased to reflect the fact that these are relationships based on statistical analysis, through which mechanisms can be inferred, but not proven. You do this well in the two following paragraphs.

8. Line 427. Analysis of LOVECLIM output and comparison with HadISST. Are these data fully independent? – or are any of the same data assimilated into LOVECLIM that are used in HadISST?

Minor points

Figure captions. Figure 1 is titled 'Twentieth century climate trends in the Southern Hemisphere. Panel A shows trends but panels B and C do not. Please retitle this figure. The units for panel A need to be clearer, is it trend per year, or over the entire period? (this point is valid for all plots showing trends).

Figure 3. Extend the x-axis on Figure 3. This is arguably the most important figure in the paper – so make it clearer to see.

Define CE on first use, and you have two different CE's, so you need to distinguish between them.

Line 93, change 'role (if any)' to 'potential role'

[Figure]

Line 129, data from Macquarie Island 'were'

Line 260. Change 'describe' to 'show'

---

## Author Comment (AC1) · 6 Feb 2017

[The comments in the response to reviewers is given below but the attached pdf supplement includes figures and a table.]

REVIEWER #1 Line 98: You say that Macquarie Island is highly sensitive to Rossby wave propagation, but the correlation values don't seem that high in Figure S4 ($\sim$0.3–0.4). Can you expand on this claim, or soften it?

Given the correlations stating the sensitivity is 'high' is probably too strong. However, numerous studies have identified the region around Macquarie Island as sensitive to Rossby wave propogation and we have cited these here to support our statement. The

text now reads: '. . .and are sensitive to Rossby wave propagation from the tropics to the high-latitudes (Adamson et al., 1988; Ding et al., 2012) (Fig. S4).'

Section 2.1: This section is a bit unclear. What quality control has been done to the AAE data? Where are they available? You also mention comparing the satellite SSTs to the AAE reports, but was SST recorded during 1912–1915? Is this the Buckles Bay measurements, or are they the Bureau of Meteorology records? A version of Table S1 here might make this section more clear.

The AAE data were obtained from Newman (1929). We apologise over the ambiguity of the statement regarding the SST measurements made during the AAE. To clarify, we have rephrased the source of the data: 'The daily and monthly meteorological data from Macquarie Island was obtained from the reduced AAE dataset (Newman, 1929) and since 1948, the Bureau of Meteorology (http://www.bom.gov.au/climate/data-services/). Twice-daily SST measurements were also taken from Buckles Bay during the AAE, with subsequent observations made again during the 1950s and 1960s (Loewe, 1968, 1957);' We have also added a statement in Table S1 caption on where to obtained the original AAE data: 'The early twentieth century AAE data from Macquarie Island (1912-1915) can be obtained from Newman (1929).'

Section 2.2. This section (and Tables S2 and S3) confused me. What are you trying to show? That the interannual variability of the AAE data is within the modern range? That there is a temperature trend? What are the statistical results relative to? Figure 3: Do you have a possible explanation for the disagreement between the obs and the reconstruction in the 1950s?

This was somewhat ambiguous(!) and apologise for any confusion. We have rephrased the main text to clarify the purpose of these preliminary analyses: 'To extend the satellite record for the southwest Pacific, we focused on the subantarctic Macquarie and Campbell islands. For comparison to the AAE 1912-1915 record, modern-day atmosphere-ocean Macquarie Island temperature measurements were compared in

four-year bins (Tables S2 and S3). The interannual variability in the most complete dataset (that from Macquarie Island) is relatively large. Student's t-tests (two-tailed) of the four-year average monthly data relative to 1912-1915 indicate that the most consistently warmer conditions are during February-April (Tables S2 and S3). This analysis illustrates a trend towards seasonally-restricted warming only during the late austral summer and autumn. Intriguingly, no pervasive warming is observed across the austral spring and most of the summer when ENSO and SAM are today known to play a dominant role on regional climate variability (Ciasto and Thompson, 2008).' We have also revised the captions on Tables S2 and S3 to make the comparison to the AAE period clearer: 'Table S2: Four-year binned monthly atmospheric temperatures for the Isthmus, Macquarie Island. Statistically-significant two-tailed t-tests differences between the modern record and the original Australasian Antarctic Expedition (AAE) are given in bold: *$p<0.05$ and † $p<0.01$.' and 'Table S3: Four-year binned monthly sea surface temperature (SST) for Buckles Bay, Macquarie Island. Statistically-significant two-tailed t-differences between the SSTs obtained during the 1950s and 1960s (Loewe, 1968, 1957) and MODIS 4 km-resolved 11 $\mu$m daytime satellite observations to the original Australasian Antarctic Expedition (AAE) are given in bold: *$p<0.05$. Note, the 1950s and 1960s SSTs obtained from Macquarie Island are reported as monthly averages by Loewe (1957, 1968), precluding t-tests.' Unfortunately, few if any, proxy methods are absolute and it does appear for fours years, the temperature reconstruction does diverge from the absolute measurements. One possibility may be the teleconnection between Campbell Island (where the trees were obtained) and Macquarie Island (where the comparison is least favourable). However, taken over the full observational record, the most southerly growing trees in the southwest Pacific appear to be significantly controlled by temperatures averaged over the spring and summer months (October to March) (Table S4).

Section 3.2. To me this section makes more sense at the start of section 2.3, as it does not add much here.

Absolutely. Change made.

Section 3.3. From where does HadISST get its data? Given the data scarcity in these high latitudes, it would be worth confirming that HadISST does not have the same data as Loewe, or has not approximated the Bureau's air temperatures as SSTs, given the high level of agreement between the two.

We have checked HadISST using ICOADS and find no record of a contribution from the Loewe datasets. However, it should be noted that the increased expression of the 2.4 and 3.1 year periodicities in the extracted HadISST data commences from the mid-twentieth century and through into the satellite era (i.e. even when there are no direct SST observations at Macquarie Island), providing confidence in the identification of increased variability.

Section 3.3 as well. Chelton and Raisin (2016) mention that there is an increase in standard deviation in HadISST from 1949, at least over the North Pacific, due to artifacts in the dataset. It would be worth acknowledging this or even better, arguing that your dendrochronology results show a real variance increase that is independent of this artifact in the observational data.

This is an excellent point. Thank you. We have revised the text using a variation of the words suggested by the reviewer: 'Recent work by Chelton and Risien (2016) suggest that there is an increase in standard deviation in HadISST from 1949. Our tree-ring temperature reconstruction, however, shows a real variance increase that is independent of this artifact in the observational data.'

Section 3.4. This is a nice additional independent line of evidence, I look forward to reading the future research.

We thank the reviewer for their kind and supportive words.

Line 376–380. Historical wind data are notoriously rubbish (e.g. Jones et al. 1997; Jakob, 2010). I'm suspicious of drawing any conclusion on AAE wind data unless

you have good metadata about where exactly the wine vane was located. Jakob D. 2010. Challenges in developing a high-quality surface wind-speed data-set for Australia. Australian Meteorological and Oceanographic Journal 60: 227–236. Gallego D, Garcia-Herrera R, Calvo N, Ribera P. 2007. A new meteorological record for Cadiz (Spain) 1806–1852: Implications for climatic reconstructions. Journal of Geophysical Research 112, DOI:10.1029/2007jd008517.

This is a fair point. The meteorological station established by the Australasian Antarctic Expedition during 1912-1915 is close to the location of the current station. However, differences almost certainly exist. As a result, we have replotted the wind data as seasonal averages (rather than the 30 years used in the submission); unfortunately the wind data from Campbell Island is only available from the 1990s so is not plotted here. In the text we have toned down the conclusions possible for comparison to the AAE average. The text now reads as '…we do observe a marked increase in wind strength across the late twentieth century, with a long-term intensification (with high variability) of winds that closely parallels air temperatures over Macquarie Island (Fig. 5D). The original AAE data is plotted for completeness but given uncertainties over the reliability of historic observations (Jakob, 2010) a direct comparison is questionable. This trend towards stronger winds is accompanied by an increase in sunshine hours over Macquarie Island (Fig. S9), consistent with reduced cloud cover, but any associated increase in sensible heat flux appears to be substantially modulated by increased airflow over cooler surface waters in the southwest Pacific (Thompson et al., 2011) (Fig. S1). Our results, therefore, are in line with the observed (post-1979) spring-summer trend towards windier conditions in the southwest Pacific (Fig. S1).'

Lines 444–445: You say that "The close similarity between the LOVECLIM output and HadISST argues against any bias in the latter dataset for this region", however you have forced LOVECLIM with HadCRUT3. As I understand it, HadCRUT3 gets its SST data from HadSST2, which in turn gets its data from ICOADS. HadISST also uses COADS data (the precursor to ICOADS) to increase data coverage, although I'm not

sure exactly where. You need to check the data sources for the region for both datasets to claim that these results are independent.

This is absolutely correct and thank the reviewer for pointing this out. We have removed the statement to this effect so that the comparison of the datasets is restricted to the following statement: 'Parallel changes in SST magnitude and trend in the southwest Pacific using both the LOVECLIM model output and HadISST (Rayner et al., 2003) is consistent with our reconstruction of subantarctic island temperatures (Figs. 3, 8 and 9).'

Technical corrections Line 10: I would reverse this sentence to improve readability: "SAM and ENSO play a dominant role in this as modes of large scale variability". Text changed. Line 129: Macquarie Island were, not was. Text changed. Lines 197 and 198: I think you mean Figure S11 and S12. Sorry, yes. Text changed. Line 263: I'd put a comma after limiting, the sentence is a bit confusing otherwise. Line 289: add a 'to' after the word limited Text changed. Line 359: data are Text changed. Figure 1, and Figures S1–S3: What is the source of the data fields you are plotting? I'd also add °C to the colour bar just for clarity. We have changed the figure accordingly. We have also changed the caption to read: 'Figure 1. Ocean-atmosphere coupling in the Southern Hemisphere. A. Significant (pÂǎ< 0.05) austral sea surface temperature (SST ËŽC/decade; shading) and 925-hPa winds (vectors) trends for December-February since 1979. Temperatures based on SSTs from the HadISST dataset (Rayner et al., 2003); winds from ERA Interim (Dee et al., 2011)...' ËŽC/decade has been added to the figure for clarification as the reviewer suggests. Figure 1: South Georgia is labeled as SG in the image, but GI in the caption. Text changed. Figure 2: Consider using dotted/dashed lines for colour-blind readers. We have made some modifications (dashed the raw data curve and made the line for the EPS black and a smaller stroke) which we hope helps. Figure 3: I would make the orange block lighter to make the graph easier to look at. Figure changed as suggested. We have also changed Figure 9 to the same colour. Figure 4: Consider changing the colours here to grey and black for colour-blind

readers. Figure changed as suggested. We have dashed the grey line to make it easier to see. Figure S4: Which areas are significant? Those that are shaded? Please clarify. You could even add the significance threshold correlation value. The detail was lost a little in the caption. We have rephrased to the following: 'Statistically significant spatial correlations between detrended and deseasonalised Nino 3 sea surface temperature (pfieldÂă< 0.05) (Rayner et al., 2003) and 850 hPa height anomalies...' We hope this is clearer. Tables S5 and S6 could be swapped so they are in the order in which they are mentioned. This goes for the figures and supplementary figures as well. At times I got a little lost because the Figure references were not in order (e.g. Figures 8 and 9 mentioned before Figures 5–7). We must apologise for this. We have swapped the order of the supplementary table. Figures 8 and 9 were incorrectly cited in the methods section which has now been removed.

REVIEWER #2

General comments This is a very interesting and carefully written manuscript that sheds significant light on pre-instrumental climate variability over the southern oceans. The evidence for an increase in climate variability is compelling, as are the links to tropical/ENSO variability and Rossby wave propagation. There is quite some overlap between sections 2 and 3 and there is some repetition. Can you rearrange the text to keep the results confined to section 3? We thank the reviewer for their positive comments but apologise for overlap in sections 2 and 3. We have put what was Section 3.2 describing the tree-ring methods into Section 2.3 (also requested above).

Specific comments Lines 59-60: Could also add Jones et al. (2016) here. Citation added.

Line 84: I wouldn't use the word "competing" here. Both flows act to reduce the north-south temperature gradient, hence are really working in the same sense. This is a fair point. We have removed the word 'competing'.

Technical corrections Line 211: Define NCEP. NCEP defined. McGlone et al (2010)

reference is missing an author. Author James Renwick added to McGlone et al. (2010).

REVIEWER 3

I have reviewed a previous version of this paper before for another journal. I am pleased that the authors have addressed many of my previous concerns, including taking up my suggestion to link temperatures at the subantarctic Islands to local sea surface temperatures, as well as to tropical climate variability.

We fear there may have been a misunderstanding over the reviewer's interpretation of a previous draft of the manuscript. We thank the reviewer for their support and glad they like the revisions.

I therefore recommend publication subject to major revisions. I apologise that this review is late.

1. Make sure that all figures, both in the main text and supplementary information, are numbered in order of discussion in the text, this is not done for all figures.

We have identified the misnumbered tables and figures. This has now been corrected (see above). Apologies for the confusion.

2. Section 3.1. Modern climate changes. Figure 1c shows well the coupling between atmospheric and oceanic temperatures at Macquarie Island over the annual cycle. Similarly Figure 5C shows, as well as the influence of SSTs in the Nino 3 region, strong correlations with local SSTs. Given this close coupling, the influence of this strong coupling on interannual temperature variability on Macquarie Island (MI) and Campbell Island (CI) needs to be explored further in the paper, and consideration of the relative proportions of temperature variance explained by local vs remote SSTs is needed. You should therefore include the timeseries of MODIS and HadISST temperatures for gridboxes/pixels closest to the islands in Figure 1.

I would like to see this for both MI and CI. You could add the CI plots to the supplementary info, although additional panels in Figure 1 would also work. In Table 1, please

include correlations with local SSTs (Modis and HadISST). I realise the MODIS correlations will be for a shorter period, but I think this analysis would still add to the paper. This analysis will allow quantification of the relative influence of local versus remote SSTs. I realise that SSTs in these two regions may not be completely independent, so perhaps also include a column correlating NINO 1+2/3 and SSTs local to MI/CI too. I would also like to see (perhaps as a panel in Figure 9), the tree ring reconstructions with HadISST SSTs for the MI/CI region.

The Macquarie and Campbell Islands are sensitive to sea surface temperatures across the broader region. In addition to the significant relationship between the air temperatures over the two islands cited in the original manuscript (detrended July-June correlation 0.801, p < 0.0001), we have produced a new supplementary figure showing the relationship between air temperatures over the Dracophyllum growing season (October-March) for both islands and compared to SSTs extracted from both HadISST and Reynolds v2, both limited to the satellite era (1979-2014 and 1982-2014 respectively). The supplementary figure and caption is provided in the attached document.

Figure S5: Spatial correlations between detrended and deseasonalised Macquarie Island and Campbell Island mean monthly atmospheric temperatures (October-March) and sea surface temperature obtained using HadISST for the period 1979-2013 (Rayner et al., 2003) (Panels A and C respectively) and Reynolds v2 for the period 1982-2013 (Smith and Reynolds, 2005) (Panels B and D) (pfieldÂă< 0.05). The southwest Pacific (SW Pacific; 50-60ËŽS, 150-170ËŽE) and Nino 3 regions also shown.

As can be seen, regardless of the island or the dataset, the spatial correlations are the same, demonstrating a tight ocean-atmospheric coupling to regional conditions which as our study shows in turn is influenced by equatorial temperatures. As a result we have also added extra text in the Introduction to justify the study of both islands: 'The time series of observed temperatures on the two islands are highly correlated in the modern record (detrended July-June correlation 0.801, p < 0.0001) and display the same significant spatial correlation fields to regional and Pacific-wide SSTs (Fig. S5),

demonstrating a comparable climate regime.' We feel Figure 1 effectively captures the spatial and local trends in climate and SSTs, and the sensitivity of the island air temperatures to local SSTs. The subsequent analyses and figures then go on to establish the relationship to the broader region (supported by the other figures and analyses requested by the reviewers). We worry the inclusion of time series from the southwest Pacific risks making the figure confusing to the reader. We hope the extracted HadISST temperatures from the southwest Pacific in Figure 9 will suffice to further reassure the reviewer of the tight regional ocean-atmospheric coupling demonstrated by our work.

We have also undertaken analysis comparing the significance of the correlation between the island air temperatures and the regional SSTs and included a new column in Table 1. The new text in the Discussion has been added: 'A similar positive correlation to spring-summer SSTs is observed with both Macquarie Island (Fig. 5C) and Campbell Island (Fig. S5) with highly significant relationships to a sector in the southwest Pacific (50ËŽ-60ËŽS, 150ËŽ-170ËŽE; Table 1), supporting our earlier observation of the thermal coupling between atmospheric and ocean temperatures but extending across the broader region.' The revised Table 1 is shown in the attached response.

Table 1: Correlations and significance of relationship between subantarctic island air temperatures and measures of regional (50-60ËŽS, 150-170ËŽE) and equatorial Pacific sea surface temperature (SST) and atmospheric circulation. Regional and Nino temperature anomalies as calculated from HadISST (Rayner et al., 2003); the Southern Oscillation Index (SOI) as reported by Ropelewski and Jones (1987). Deseasonalised and detrended correlations derived for the period CE 1979 to 2014. Significance indicated as follows: *p<0.05, † p<0.01, and ‡ p<0.001.

Given the shortness of the record we have not attempted spatial correlations between our data to MODIS. Instead, we feel it is more important to demonstrate the spatial correlations between the subantarctic islands and regional and Pacific-wide temperatures is a robust relationship, regardless of the way in situ and satellite observations have been interpolated (Kennedy, J. J.: A review of uncertainty in in situ measurements and data sets of sea surface temperature, Reviews of Geophysics, 52, 1-32, 10.1002/2013RG000434, 2014). The correlations reported in the previous manuscript were all restricted to the satellite era (post-1979) to provide the greatest confidence in the spatial and temporal relationships. The new results generated using the Campbell Island air temperatures and the Reynolds v2 SSTs (1982-2014) also has the additional benefit of supporting the interpretation of the HadISST and Reynolds Hovmöller plots (now Figure S14). Also we feel any attempt to quantify the relative regional versus equatorial Pacific influences on island temperatures may lead to a misleading interpretation. Ultimately, we hope our work has shown local temperatures are not independent of remote temperatures, and that equatorial conditions play a significant role in modulating local SSTs (and by association. air temperatures over the subantarctic islands).

3. The penguin and seal analysis is not convincing and I recommend either removing this from the paper, or toning this section down. Although I am not an expert in links between penguin populations and climate, I am not convinced that the coincidence of the decline with an increase in climate variability does prove a causal linkage.

Additionally, from reading one of the sources cited (Morrison et al. 2015), these authors state that although there has been an overall long-term decline, Rockhopper Penguin numbers on Campbell Island have recovered/stabilised over the period 1996-2012 (a fact not explicitly mentioned in the text, rather hinted at by stating that 94% of the decline had occurred by the mid 1980s). Morrison et al. state 'the recent decline occurred during the period 1984–1996 and was followed by overall population growth 1996–2012, concurrent with lower SSTs and an increased abundance of a key prey species'. This would seem to fit with the SST changes discussed in Point 1 above, rather than to temperature variability.

We are not providing absolute proof that the increased climate variability is the cause of mammal and sea bird population decline in the southwest Pacific. However, this is a considerable area of uncertainty with no factor(s) yet clearly identified as the cause of the order of magnitude decline of populations of multiple species across the region;

a process that began sometime from the 1940s. We feel it would be remiss to not describe what is a major biological change in the region, particularly given the parallel change in increased climate variability identified here and population decline. As Reviewer 1 points out, this work provides an independent line of evidence to our findings. Importantly, the reviewer appears to have misunderstood the study by Morrison et al., 2015. Whilst a part of the text may state a modest growth of RockhopperÂǎnumbers on Campbell Island compared to the penultimate study period (based on analysis of photographs), overall the population has suffered an overall (net) ~1.5% decline since the mid-1980s, suggesting no recovery and consistent with a negative impact from pervasive high climate variability. We have, however, provided further details (including potential land use impacts on the islands) and toned down any implication that this is a definitive study. The following text now comprises Section 3.3 on 'Marine population changes':

[revised manuscript text omitted]

4. Clarity of phrasing. - First sentence of the paper. Figures 1 and S1 do not show ice sheet dynamics. The Jones et al. (2016) provides a recent summary of the key trends across the mid to high southern latitudes. We have moved reference to Figures 1 and S1 so that they immediately follow '...variability in atmospheric and ocean temperature and circulation'. - Line 68. 'Late twentieth century Southern Ocean climate' – I assume you mean climate over the Southern Ocean region, not the climate of the ocean itself. We have changed the text to: 'Late twentieth century climate over the Southern Ocean...' - Line 63, make it clear what 'This' is referring to. We have changed the text to: 'The above uncertainties are...' - Page 2 line 75. 'ENSO is associated with spatially different temperature and wind trends across mid to high latitudes (Figure S3). Figure S3 shows the regression of Nino 3 and 3.4 on SST and wind - therefore you mean the signal/influence, not the trend. Yes, thank you. We have changed to 'relationships'.

- Section 2.2. line 150, comparison of AAE temperatures to those on the islands. Firstly, make it clear in the text that it's temperature measurements. You state that 'this analysis illustrates the trend through time rather than interpreting specific comparisons'. I do not understand what you mean by this, please rephrase. We have rephrased to make clearer. The paragpraph now reads: 'To extend the satellite record for the southwest Pacific, we focused on the subantarctic Macquarie and Campbell islands. For comparison to the AAE 1912-1915 record, modern-day Macquarie Island temperature measurements were compared in four-year bins (Tables S2 and S3). The interannual variability in the most complete dataset (that from Macquarie Island) is relatively large. Student's t-tests (two-tailed) of the four-year average monthly data relative to 1912-1915 indicate that the most consistently warmer conditions are during February-April (Tables S2 and S3). While we are aware of the issues surrounding

multiple-testing, this analysis illustrates a trend towards seasonally-restricted warming only during the late austral summer and autumn. Intriguingly, no pervasive warming is observed across the austral spring and most of the summer when ENSO and SAM are today known to play a dominant role on regional climate variability (Ciasto and Thompson, 2008).'

- Line 291. You state 'a long-term trend towards increasing temperatures from the 1960s that reached a maximum during the late 1980's. I wouldn't call a 20 year trend a long-term trend, and temperatures reach a maximum in the late 1980s, not the trend. It would also help the reader if you mark 1912-15 on Figure 3. You then state that the late 1980s were warmer compared to 1912-1915, indicating some kind of linear behaviour – but what Figure 3 shows to me is lots of variability, and there are periods since 1912-15 that are cooler, and others that are warmer. Please rephrase this more carefully and clearly. We have rephrased. The new text reads: 'We find highly-variable growing season (spring-summer) temperatures that parallel meteorological observations on the subantarctic islands for the period of overlap (including the original AAE) (Fig. 3), with a trend towards increasing temperatures from the 1960s that reached a maximum during the late 1980s (∼1ẼŽC warmer on Macquarie Island compared to period 1912-1915).' The observations from the original AAE are shown in Figure 3B.

- Line 481, 'contemporary equatorial Pacific temperatures may now be a permanent feature across the mid to high latitudes' – do you mean the influence of contemporary equatorial Pacific temperatures. This sentence currently doesn't make sense. This is correct. We have changed the text to read: 'Our findings, however, provide a long-term perspective that suggests modern observed high interannual variability was established across the 1940s, and that the influence of contemporary equatorial Pacific temperatures may now be a permanent feature across the mid to high latitudes.'

- Caption of Fig S7, you state that there is no sustained change in wind direction since the expedition. I do not think that this can be concluded from this plot. Please rephrase (our just delete). See point 6 below. We have deleted this statement.

Discussion 5. An overall comment to the discussion section is that I suggest having sub-headings in this section, to guide the reader, as well as addressing the issue of order of figure numbering, as currently this section is a little difficult to follow. You may wish to consider instead of having separate results and discussion sections, restricting to having an overall 'Results and Discussion' section (as there is a lot of analysis in the results section), but with clear subheadings that lead the reader through the analysis. This is a good idea and we have changed the structure of the manuscript with a 'Results and Discussion' section to more clearly describe our findings.

6. Line 377. You state there's no long term trend in solar radiation – it appears from Figure S8 that there has been a shift towards greater sunshine hours since the mid-1960s. This needs exploring/explaining. Also in this paragraph, you state that there has been 'a long term intensification' of winds. Please remove the phrase 'long-term intensification'. What can be concluded from Figure 5D is that winds since 1950 are stronger than those during 1912-1915, and that the 1981-2010 winds are stronger than the 1951-1980. This could be evidence of a long-term intensification, but given the strong interannual variability in this region, there does need to be the caveat that the 1912-1915 winds are a snapshot of winds in a region with strong variability. Also in this paragraph, Line 380/discussion of Figure S1. It needs to be shown on Figure S1 which trends are significant, perhaps marking these with a different colour arrow. Also, marking of MI and CI on this plot would be useful. We have modified the discussion of wind strength changes and sunshine hours in line with Reviewer 1. The new text reads: 'Although we find no evidence for a sustained shift in airflow direction that parallels the observed trend in subantarctic temperatures (Fig. S8) we do observe a marked increase in wind strength across the late twentieth century, with a long-term intensification (with high variability) of winds that closely parallels air temperatures over Macquarie Island (Fig. 5D); the original AAE data is plotted for completeness but given uncertainties over the reliability of historic observations (Jakob, 2010) a direct comparison is not possible. This trend towards stronger winds is accompanied by an increase in sunshine hours over Macquarie Island (Fig. S9), consistent with reduced cloud

cover, but any associated increase in sensible heat flux appears to be substantially modulated by increased airflow over cooler surface waters in the southwest Pacific (Thompson et al., 2011) (Fig. S1). Our results, therefore, are in line with the observed (post-1979) spring-summer trend towards windier conditions in the southwest Pacific (Fig. S1).' We have also modified Figure S1 with only those trends that are significant at p<0.05.

7. Line 394. You state that you 'observe a Rossby wave train', and online 396, that 'We find that post-1979 warmer temperatures in the Nino 3 region leads to deep convection. . .. Forcing an atmospheric wave train'. The discussion in this paragraph needs to be rephrased to reflect the fact that these are relationships based on statistical analysis, through which mechanisms can be inferred, but not proven. You do this well in the two following paragraphs. We have revised the text to the following: 'We observe what appears to be a Rossby wave train similar to the PSA climate mode of variability during the austral spring-summer (Ding et al., 2012; Mo and Higgins, 1998; Trenberth et al., 1998). We find that post-1979, warmer temperatures in the Nino 3 region leads to deep convection and upper-level divergence flow (at 300 hPa) (Fogt et al., 2012; Ding et al., 2012; Trenberth et al., 1998) (Fig. S16), apparently forcing an atmospheric Rossby wave train southeast into the extratropics manifested as cyclonic anomalies south of New Zealand – consistent with the relationship observed with Macquarie Island temperatures (Fig. 5) – that extend across the Pacific as anticyclonic anomalies in the Amundsen-Bellingshausen seas and cyclonic anomalies off the east coast of South America (Ciasto and Thompson, 2008; Mo and Higgins, 1998).'

8. Line 427. Analysis of LOVECLIM output and comparison with HadISST. Are these data fully independent? – or are any of the same data assimilated into LOVECLIM that are used in HadISST? The reviewer is correct. See the discussion point above in response to Reviewer 1. As a result we have also modified the related text to: 'Over the past century, we find increasingly stronger westerly winds across the Southern Ocean with a marked intensification in the southwest Pacific and Antarctic Peninsula during

the most recent decades with more easterly airflow over the Ross Sea (Fig. 8C), trends also observed in estimates derived from the ERA Interim dataset (Fig. 8D) (Dee et al., 2011), and consistent with the observational record from Macquarie Island (Figs. 5D).'

Minor points Figure captions. Figure 1 is titled 'Twentieth century climate trends in the Southern Hemisphere. Panel A shows trends but panels B and C do not. Please retitle this figure. The units for panel A need to be clearer, is it trend per year, or over the entire period? (this point is valid for all plots showing trends). We agree. This is ambiguous. We have revised the text including changing the title of Figure 1: 'Figure 1. Ocean-atmosphere coupling in the Southern Hemisphere. A. Significant (pÂǎ< 0.05) austral sea surface temperature (SST ËŽC/decade; shading) and 925-hPa winds (vectors) trends for December-February since 1979. Temperatures based on SSTs from the HadISST dataset (Rayner et al., 2003); winds from ERA Interim (Dee et al., 2011)...'

Figure 3. Extend the x-axis on Figure 3. This is arguably the most important figure in the paper – so make it clearer to see. Define CE on first use, and you have two different CE's, so you need to distinguish between them. The CE is now defined in text and the x-axis of Figure 3 has been extended.

Line 93, change 'role (if any)' to 'potential role' We have changed the text as suggested.

Line 129, data from Macquarie Island 'were' Line 260. Change 'describe' to 'show' We have changed the text as suggested.

Please also note the supplement to this comment:
http://www.clim-past-discuss.net/cp-2016-114/cp-2016-114-AC1-supplement.pdf

———————————————————